# Aging Behavior of Polyethylene and Ceramics-Coated Separators under the Simulated Lithium-Ion Battery Service Compression and Temperature Field

Wang Qian [1,†], Shuqiu Wu [2,†], Caihong Lei [1,*], Ruijie Xu [1,*] and Yanjie Wang [2]

[1] School of Materials and Energy, Guangdong University of Technology, Guangzhou 510006, China; qw182625@126.com
[2] Shenzhen Senior Technology Material Co., Ltd., Shenzhen 518106, China; beyond_msst@163.com (S.W.); wangy.j@foxmail.com (Y.W.)
[*] Correspondence: lch528@gdut.edu.cn (C.L.); xu565786@163.com (R.X.)
[†] These authors contributed equally to this work.

**Abstract:** In this paper, a device was set up, which could simulate the separator environment in the battery to track the influence of compression, temperature, and the electrolyte on the structure and electrochemical performance of separators. A commercial polyethylene separator and alumina- or boehmite-coated separators were selected, and the high-temperature cyclic compression was carried out in a mixed solvent environment with a ratio of vinyl carbonate and diethyl carbonate of 1:1. Compared with that compressed for 50 cycles under room temperature, the compression at 60 °C resulted in pore structure deterioration in the polyethylene separator. The oxidative voltage limit was reduced to 3.6 V, and after 200 charge and discharge cycles, the capacity was reduced by more than 50%. For the coated separator, the presence of a coating layer exhibited some protective effects, and the microporous structure in the base membrane was preserved. The oxidative voltage limit was above 4.2 V. However, as a result of the compression, the coating particles were still inserted into the pore structure, leading to a decrease in porosity and a decrease in discharge capacity, especially at a rate of 4 C. Compared with that coated with alumina particles, the interface resistance for the separator coated with boehmite particles was minimally affected, and the electrochemical performance after cyclic compression under 60 °C was better, exhibiting higher application ability.

**Keywords:** separator; aging; cyclic compression; electrochemical performance

## 1. Introduction

Nowadays, rechargeable lithium-ion batteries (LIBs) have become the mainstream of portable electronic devices and electric vehicles [1,2]. Along with the massive use of LIBs, battery failures have become the focus of research in the industry and in the science field. On one hand, battery failures are caused by the volume expansion of positive and negative electrodes during cyclic charging and discharging [3], due to the compressive stresses generated inside the battery. On the other hand, the decomposition of the liquid electrolyte to produce gases due to rapid heating of the battery [4] can also lead to excessive internal stresses. In addition, the regrowth of the positive and negative electrode interface (SEI) membranes also generates compressive stresses inside the battery [5,6]. As one of the core components in LIBs, the separator is located between the positive and negative electrodes, which can not only avoid the internal short circuit caused by the physical contact of the positive and negative electrodes, but also ensure the migration of lithium ions (Li$^+$) through the porous structure under the excitation of the redox reaction [7,8], and the interconnected microporous structure in the separator determines the Li$^+$ transport path [9,10]. The compressive stresses generated during the actual use of the battery can cause the deformation of the microporous structure, thus affecting the ion transport and

the overall performance of the battery [9]. However, the separator not only encounters compressive stresses under the operating state of lithium-ion batteries, but also faces high temperatures and an electrolyte environment [11]. The separator's aging can directly lead to the deterioration of battery performance and even battery failure. Therefore, tracking the microstructural changes of the microporous separator during battery use is important in the study of battery safety [12,13].

The separator mainly faces mechanical stress, high temperatures, and the swelling and softening of the liquid electrolyte during battery service [14–19]. However, due to the resemblance of the battery to a "black box" [20] after assembly, it is impossible to directly observe and characterize the microstructural changes of the separator during battery use. Wang et al. [21] explored the physical properties of the separator by disassembling batteries under a variety of common battery abuse conditions. However, the disassembly experiment of the battery could not track the microstructural changes of the separator during use in situ. During battery service, active material particles, redox reaction by-products, and lithium metal precipitation [22] hinders the pores and input cavity volume of the separator, so it is difficult to directly observe the microstructural changes of the separator under the coupling of the liquid electrolyte environment, mechanical compression, and high-temperature field.

Therefore, many studies have simulated the application environment of separators in batteries. Many works have followed the changes under compressive stress. Cannarella et al. [23,24] conducted an in-depth study of the compressive and tensile mechanical properties of a separator (Celgard 3501, SKJYLEAN, Suzhou, China) at different strain rates and in different fluid environments. In their study, they highlighted the anisotropic properties of the separator and the importance of the simultaneous measurement of tensile and compressive properties. Zhang et al. [25] studied the failure mode of the separator by establishing a punch device with a hemispherical punch of different diameters, which helps us further understand the failure mode of the separator during battery service. Lagadec et al. [26] characterized how the microstructural properties of the separator change with compressive strain by simulating how the separator deforms during the compression process, and simulated the effect on the microstructure and the transport of lithium ions through the separator.

However, in addition to the compressive stress, the influence of the liquid electrolyte environment and high temperature could not be ignored. Studies have shown that after battery cyclic charging and discharging, some micron-sized crater-like particles appear on the separator surface corresponding to the anode, which may be the result of local overheating, mechanical stress, and the swelling and softening of the liquid electrolyte [27]. Perea et al. [28] found that after the lithium-ion batteries were charged and discharged at a rate of 2 C, the internal temperature of the battery could reach 60–70 °C. Chen et al. [29] analyzed the AFM image of the separator and found that the separator began to appear to have closed pores at 90 °C, thereby changing the ion migration pathway. Kalnaus et al. [30] revealed that the dry method-prepared separators have a specific temperature dependence during compression, and proposed that the temperature field caused lasting changes in the mechanical behavior of the separator. Therefore, it is necessary to consider the influence of temperature when studying the cyclic compression of the separator. In addition, during the actual operation of the battery, the electrolyte is always in the battery, and the separator expands and softens in it [31]. Therefore, in exploring the compression behavior of separators during battery service, it is necessary to consider the coupling effect of the temperature field and liquid electrolyte.

In our previous work [20,32], one setup was first designed by combining the punch test periodically and the swelling and softening of separators in electrolyte solvents. Here, the high temperature's influence was further included, and the compression behavior and corresponding electrochemical performance of the commercial polyethylene microporous separator and the alumina- or boehmite-coated separators in an electrolyte environment at 60 °C were explored by simulating the battery working environment, and the influence

of microporous structural changes in the ion conduction during the aging process of the commercial separator was clarified.

## 2. Materials and Experimental Procedure

### 2.1. Materials

Three commercial separators were supplied by Shenzhen Senior Technology Material Co., Ltd., Shenzhen, China, including a commercial wet-processed polyethylene separator (W-PE, with a thickness of 9 μm) and coated separators with alumina or boehmite on one side (W-PE-A and W-PE-B, with thicknesses of 9 + 3 μm). The liquid electrolyte (1 M LiPF$_6$ in EC/DEC, with a volume ratio of 1:1), lithium iron phosphate cathode (ratio of active substance: 91.5 wt%, collector electrodes carbon-coated Al, and rolled electrodes), and lithium sheets were purchased from Guangdong Canrd New Energy Technology Co., Ltd., Dongguan, China. EC (ethylene carbonate) and DEC (diethyl carbonate) were purchased from Guangzhou Sopo Biological Technology Co., Ltd., Guangzhou, China.

### 2.2. Experimental Setup and Sample Preparation

The experimental device for separator compression deformation in an electrolyte is shown in Figure 1. A circular ring fixture (with an inside diameter of 40 mm) holds 10 layers of separators on the electrolyte solvent cup holder. The hemispherical indenter (customized stainless steel) with a diameter of 30 mm was subjected to reciprocating compression under the control of a universal testing machine (Inspekt Blue 5 kN, Hegewald-Peschke, Saxony, Germany).

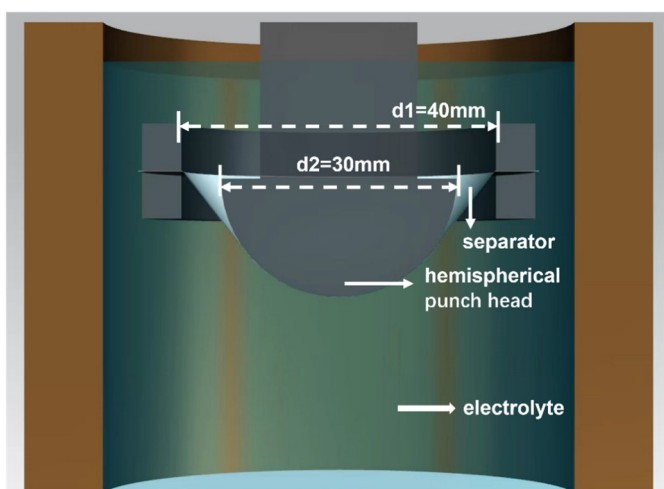

**Figure 1.** Cross-sectional view of the experimental facility for cyclic compression.

Before the test, the W-PE, W-PE-A, and W-PE-B separators were cut into squares with a side length of 45 mm and fixed in the test container. EC and DEC were mixed at a ratio of 1:1, and then poured into the container with a separator sample to simulate the solvent environments in the battery. The experimental device was pushed into a 60 °C oven, and we let it stand for 20 min to ensure that the separator was completely wetted and the solution temperature reached 60 °C. Then, 50 cyclic compression tests were performed at 60 °C and at room temperature, with a compression speed of 10 mm/min and a punch displacement set at 20 mm, respectively.

### 2.3. Testing and Characterization Methods

The microstructure of the separator surface was characterized by field emission scanning electron microscopy (FESEM, HITACHI SU8010, HITACHI, Tokyo, Japan). All separators were scanned and tested after 60 s of platinum ion sputtering under a vacuum. For the coated samples, W-PE-A and W-PE-B, after cyclic compression, the surface particles

were washed away prior to SEM measurement. For all other measurements after the cyclic compression, this washing was not performed. The pore size distribution curves were obtained by the Nano Measurer system based on counting the sizes of 200 pores on the SEM graphs.

Contact angle was tested under room temperature by the continuous titration method using a dynamic contact angle meter (HARKE-SPCAX3-1, Beijing, China). The angle between the shape of the liquid/gas interface was marked on the angle between the line and the surface of the separator, and the wettability of the separator with the liquid electrolyte was judged by the size of contact angle. The liquid electrolyte was 1 M LiPF$_6$.

The permeability of the separators was determined by means of a Gurley precision instrument (4110 N TROY, Gurlery Pecision Instruments Troy, NY, USA). During the experiment, samples of various types of separators were placed in the Gurley precision instrument and tested under preset pressure conditions. The test results are expressed as Gurley values, with lower values representing better separator permeability. By comparing and analyzing the Gurley values of different separators, we could reveal the difference in porosity between different types of separators.

To evaluate the electrolyte uptake of the separator, a rigorously designed test method was used. The test procedure was carried out in an argon-filled glove box. Firstly, we accurately measured the mass of the separator's initial dry separator using an electronic balance (ME204/02) and recorded this data as $W_{dry}$ (the typical range of values for $W_{dry}$ for each of W-PE, W-PE-A and W-PE-B are 4.27 mg, 4.93 mg, 4.78 mg). Subsequently, to ensure the accuracy of the experiment, the separator was put in the liquid electrolyte (1 M LiPF$_6$ in EC/DEC, with a volume ratio of 1:1). After the separator was immersed in the electrolyte for 20 min, the separator was removed until no electrolyte dripped down, and its weight was recorded as $W_{wet}$. Finally, according to the following Formula (1), we could derive the electrolyte absorption rate of the separator:

$$\text{Electrolyte uptake } (\%) = \frac{W_{wet} - W_{dry}}{W_{dry}} \times 100\% \tag{1}$$

In order to obtain more reliable data, we needed to repeat this step 10 times. Then, the average absorption rate values were obtained.

The physical properties test data of the three separators are shown in Table 1.

**Table 1.** The physical properties of three separators.

| Sample | Thickness (μm) | Contact Angle (°) | Electrolyte Uptake (%) | The Gurley Value (s/100 mL) |
|--------|----------------|-------------------|------------------------|------------------------------|
| W-PE | 9 | 26.3 | 226 | 178 |
| W-PE-A | 9 + 3 | 23.8 | 308 | 213 |
| W-PE-B | 9 + 3 | 19.7 | 335 | 168 |

In this experiment, we used an electrochemical workstation (VMP3B-10, 10 A/20 V, Bio-Logic Science Instruments Inc., Seyssinet-Pariset, France) to measure ionic conductivity and interfacial impedance. The measurements were performed in electrochemical impedance spectroscopy (EIS) mode, which can be widely used in the analysis of various electrochemical systems. In our experiments, we set the frequency range from 10 MHz to 1 MHz, and the disturbance voltage amplitude to 5 mV. For the measurement of interfacial impedance, we used a cell assembled with lithium electrodes (lithium sheet/separator/lithium sheet). On the other hand, the measurement of ionic conductivity relied on a cell assembled with stainless steel electrodes (SS/separator/SS). Among them, due to the selection of CR2032 battery shell assembly, the diameter of the lithium sheet was 16 mm, the diameter of the separator was 20 mm, and the diameter of the steel sheet was 18 mm. The same volume of liquid electrolyte (50 μL, 1 M LiPF$_6$ in EC/DEC, with a volume ratio of 1:1) was used

during each battery assembly. During the experiments, we calculated the ionic conductivity ($\sigma$) according to the following Formula (2) [32]:

$$\sigma = \frac{L_0}{R_b \times S} \tag{2}$$

where $L_0$ and $R_b$ are the thickness (cm) and bulk resistance (ohm) of the separator, and S is the effective contact area (cm$^2$) of the separator with the stainless steel.

The linear voltametric scanning test method was able to be applied to evaluate the electrochemical stability of the battery, and its effectiveness has been well established. In this experiment, we used the SS/separator/Li cell assembly mode, where the working electrode was SS and the reference electrode was a Li sheet, with a voltage range of 0 to 6.0 V and a scan rate of 1 mV s$^{-1}$. The linear sweep curves for 3.0–6.0 V were covered in the article, and the full range (0–6.0 V) plots were placed in the Supplementary Material file as Figure S1. This series of experiments provided us with critical information about the stability of the cell, which helps us to understand and optimize the performance of the battery in depth.

A steel sheet/lithium sheet/separator/lithium iron phosphate system half-cell was assembled, where the negative electrode was a lithium sheet, and the positive electrode was a 14 mm diameter lithium iron phosphate electrode. The battery test cycle meter used was CT-1008-S1 (Shenzhen Xinwei Technology Co., Ltd., Shenzhen, China), with a voltage setting range of 2.50–4.20 V. The cycle performance test was conducted by charging at a current density of 0.5 C, discharging at a current density of 2 C, and cycling 200 times in this way. The multiplier performance test was also charged at a current density of 0.5 C, followed by discharging at six gradients of current density, which were 0.5 C, 1 C, 2 C, 3 C, 4 C, and 0.5 C, respectively, each of which was charged and discharged six times to characterize the multiplier performance of the battery assembled by the separator.

## 3. Results and Discussion

### 3.1. Microstructure of Three Separators before and after Cyclic Compression under 60 °C

Figure 2 shows the microstructures of the three separators before and after cyclic compression under 60 °C. Scanning electron micrographs of the uncoated surfaces of the W-PE-A and W-PE-B separators without cyclic compression are given in Figure S2 of the Supplementary Material. The initial W-PE separator shows an interconnected pore structure, whereas uniformly distributed oxide ceramic particles are observed on the surface of the W-PE-A and W-PE-B separators. In addition, pore structure appears among the accumulated ceramic particles, which would be beneficial to the electrolyte absorption and ensure that lithium ions could be transmitted in the porous structure. In Table 1, it can be seen that after ceramic particle coating, the electrolyte absorption ability is increased by 70%~100%, and the contact angle is decreased. These improvements in physical properties indicate that the coated separator could improve the ionic conductivity after assembling the battery.

After the cyclic compression of the W-PE separator at 60 °C, the original uniformly distributed microporous structure showed the phenomenon of "branching tension" caused by tensile deformation during cyclic compression, and the average pore size increased, which indicated that excessive deformation had damaged the pore structure. In addition, the Gurley value of the separator after cyclic compression was 127 (s/100 mL), which can be seen from the pore size distribution in Figure 3, which is due to the increase in average pore size. Partial rupture of the microporous structure and increased pore size increase the risk of short circuits and thermal runaway in the battery.

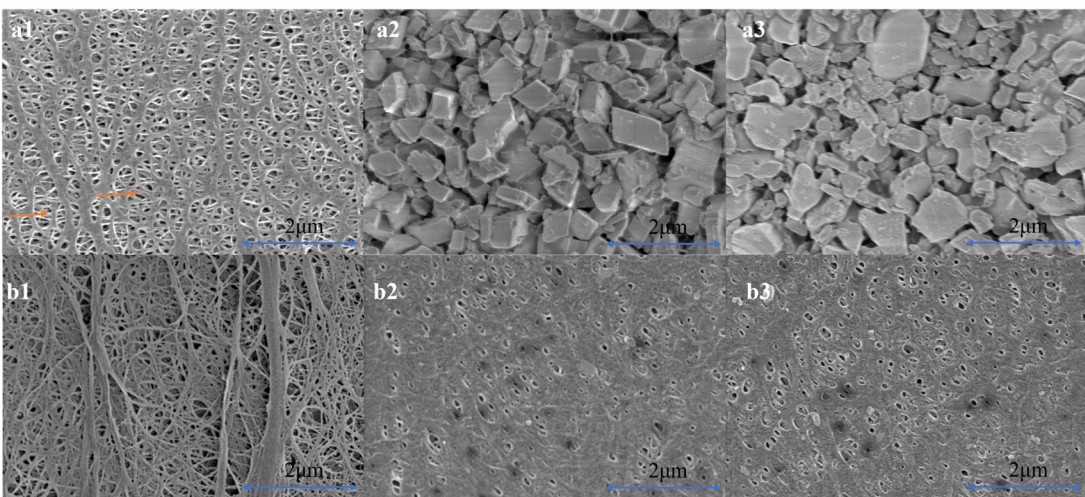

**Figure 2.** The surface SEM of W-PE (**a1**), W-PE-A (**a2**), and W-PE-B (**a3**) before cyclic compression; W-PE (**b1**), W-PE-A (**b2**), and W-PE-B (**b3**) after cyclic compression under 60 °C.

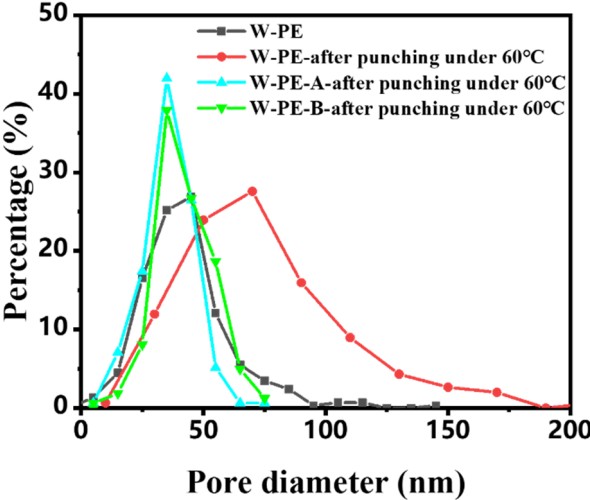

**Figure 3.** The pore size distribution of three separators after cyclic compression under 60 °C.

For the coated separator, the Gurley values of the W-PE-A and W-PE-B separators after cyclic compression were 273 and 205 (s/100 mL), respectively, indicating a decrease in porosity. As can be seen from the pore size distribution in Figure 3, the pore size of the separator after coating was below 50 nm, and the average pore size did not change significantly. The presence of the coating protects the internal microporous structure, which is the advantage of the coated separator after cyclic compression.

### 3.2. Electrochemical Performance of Three Separators before and after Cyclic Compression under 60 °C

Figure 4 gives the bulk resistance (Rb) of the three separators before and after cyclic punching under 60 °C. The high-frequency intercept on the solid axis reflects the intrinsic resistance (Rb). To obtain a more accurate calculation of the ionic conductivity, the contact and wire resistance (about 0.2–0.3 Ω) needed to be subtracted from the calculation, and then the ionic conductivity was calculated according to Equation (2).

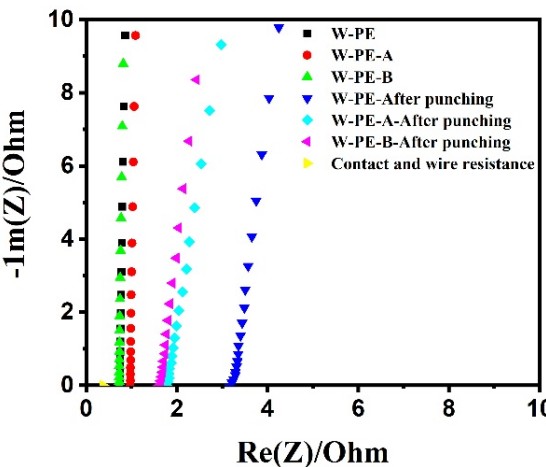

**Figure 4.** The bulk resistance (Rb) of three separators with stainless steel/separator–liquid electrolyte/stainless steel before and after cyclic compression under 60 °C.

The bulk resistance of the three separators before cyclic compression was 0.38 Ω, 0.59 Ω, and 0.34 Ω, respectively. The corresponding ionic conductivity was $1.24 \times 10^{-3}$ S cm$^{-1}$, $1.04 \times 10^{-3}$ S cm$^{-1}$, and $1.81 \times 10^{-3}$ S cm$^{-1}$, respectively. The coated separator could absorb more liquid electrolyte, provide ion pathways, and hence improve ionic conductivity. However, the ionic conductivity is also closely related to the physical properties of the separator, such as thickness, pore size distribution, and porosity. Compared with the pure polyethylene separator, the alumina coating increases the thickness, resulting in a longer transmission pathway for lithium ions. In addition, the increase in Gurley value in Table 1 for the alumina-coated separator indicates the possible blockage of alumina particles to the pores in the base membrane during coating. The corresponding bulk resistance is elevated, and the ionic conductivity is decreased compared to the pure polyethylene separator. However, for the boehmite-coated separator, due to the plate structure of boehmite particles, the pore-plugging effect is unapparent, and the higher electrolyte absorption ability increases the ionic conductivity.

After cyclic compression under 60 °C, the bulk resistance of three separators was 3.1 Ω, 1.6 Ω, and 1.4 Ω, respectively. The corresponding ionic conductivity was $1.58 \times 10^{-4}$ S cm$^{-1}$, $4.40 \times 10^{-4}$ S cm$^{-1}$, and $5.10 \times 10^{-4}$ S cm$^{-1}$, respectively. After cyclic compression, the bulk resistance increased, and the ionic conductivity was decreased by an order of magnitude. The synergistic effect of the high-temperature environment and compression stress for the uncoated separator under the working state of the battery affected the pore size distribution and pore size of the separator, thereby changing the ion transport channel and making it difficult for ions to shuttle in the separator.

After cyclic compression, the ion conductivity of the polyethylene separator was reduced by 10 times, but for that of the coated separator, only about 3 times. It can be seen from Figure 2 that the pores in the coated separator after cyclic compression are evenly distributed, and the pore size is little changed. This indicates that under the synergy of the temperature field and compressed stress, cyclic compression has the least impact on the porous structure of the separator coated with boehmite particles.

The AC impedance curves for the three separators before and after the punching experiment are given in Figure 5. The minimum interfacial resistance was approximately 268 Ω, 316 Ω, and 225 Ω for the W-PE, W-PE-A, and W-PE-B separators. Boehmite has better compatibility with organic solvents compared to alumina powders. In addition, its overall lamellar structure improves the coating flatness, which ensures that the contact between the separator coated with boehmite particles and the lithium sheet is better. This also verifies that the W-PE-B separator had the lowest interfacial resistance in the test results. However, the higher resistance of the W-PE-A separator is due to the alumina blocky particle morphology that is not conducive to interfacial contact. After cyclic compression

at 60 °C, the interfacial resistances of the three separators were 600 Ω, 530 Ω, and 447 Ω, respectively. In particular, the interfacial impedance of the W-PE separator was more than twice as high as that without cyclic compression. In general, interfacial impedance is used to characterize whether the separator and electrode are in good contact. The lower the interfacial impedance, the better the contact. The cyclic compression causes some damage to the separator structure, and this is irrecoverable. After cyclic compression, the interface resistance of the coated separator was significantly lower than that of the W-PE separator, ensuring its structural integrity and interface contact. In particular, the separator coated with boehmite had the minimum interface resistance after undergoing a cyclic compression test in the electrolyte, showing better contact with the electrode.

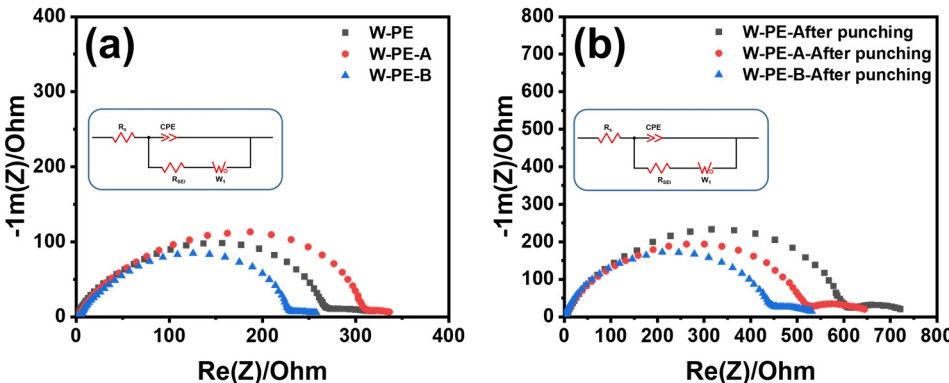

**Figure 5.** Interfacial resistance of three separators with Li/separator–liquid electrolyte/Li before cyclic compression (**a**) and after cyclic compression under 60 °C (**b**).

The linear scanning voltammetry (LSV) curves of the three separators before and after cyclic compression at 60 °C are given in Figure 6. The linear scan voltammogram reflects the electrochemical stability of the liquid electrolyte in the cell, and the relative magnitude of the onset of oxidation current in the cell before and after cyclic compression indicates the microstructural integrity of the separator. During the linear voltammograms scanning, the current of the W-PE separator before cyclic compression suddenly increased at a voltage of 4.5 V, indicating that the liquid electrolyte begins to decompose at the corresponding voltage. The battery assembled with the W-PE separator could only meet the stable charge and discharge cycle between 2.5 and 4.2 V. The onset of oxidation currents of the W-PE-A and W-PE-B separators increased to 4.7 V and 5.0 V, respectively. This supports that the coated separator not only improves the aspiration rate of the liquid electrolyte due to the presence of the coating, but also ensures the stability of the liquid electrolyte in the battery, so that the oxidative voltage limit is improved, and the battery could have charge and discharge cycles under a more stable voltage.

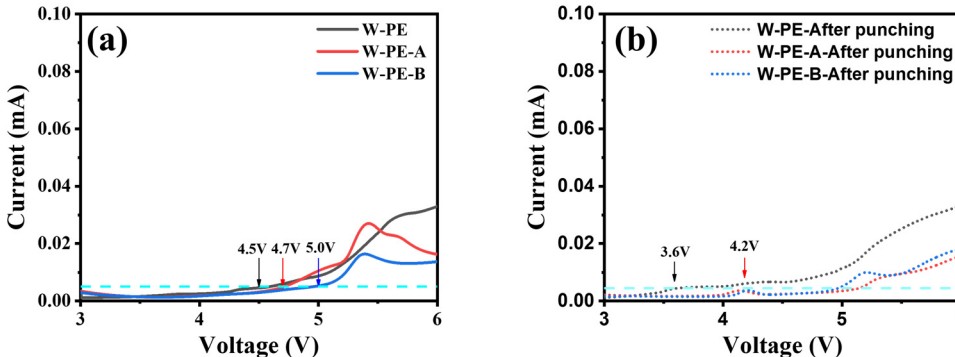

**Figure 6.** Linear scanning voltammograms of three separators using saturated 1 M LiPF$_6$ as electrolyte (**a**) before and (**b**) after cyclic compression under 60 °C.

After cyclic compression at 60 °C, the onset of oxidation current was decreased to 3.6 V, 4.2 V, and 4.2 V, respectively, indicating the change in pore structure in the separator. In particular, the maximum stable voltage of the polyethylene separator after the cyclic compression was reduced to 3.6 V. At this time, the assembled battery could not work stably at a voltage above 3.6 V, and was prone to soft short circuit, causing the battery to overheat and causing thermal runaway. The maximum stable voltage of the coated separator after cyclic compression was about 4.2 V. It could still meet the stable charge and discharge cycle within 4.2 V, so as to avoid the battery overheating caused by soft short circuiting.

The charge/discharge performance of the cells assembled using the three separators, before and after cyclic compression, was further tested. Figure 7a,c show the 0.5 C charge–2 C discharge cycle performance and C-rate performance of the three separators before cyclic compression. The capacity retention after 200 cycles of charge/discharge at 0.5–2 C for the W-PE, W-PE-A, and W-PE-B separators was 76.3%, 89.1%, and 89.4%, respectively. It is worth noting that the absorption rate increased due to the presence of the coating. In the long-cycle charge/discharge process, the capacity of the coated separator decreased slower than that of the polyethylene separator, because the lithium ions in the electrolyte were consumed during the battery cycle to form a stable SEI membrane. As a result of the improved absorption rate of the coated separator, the electrochemical stability was also improved, as can be seen from the LSV curves, and the coated separator had a nearly 13% higher capacity retention than the W-PE separator after 200 cycles of the assembled battery.

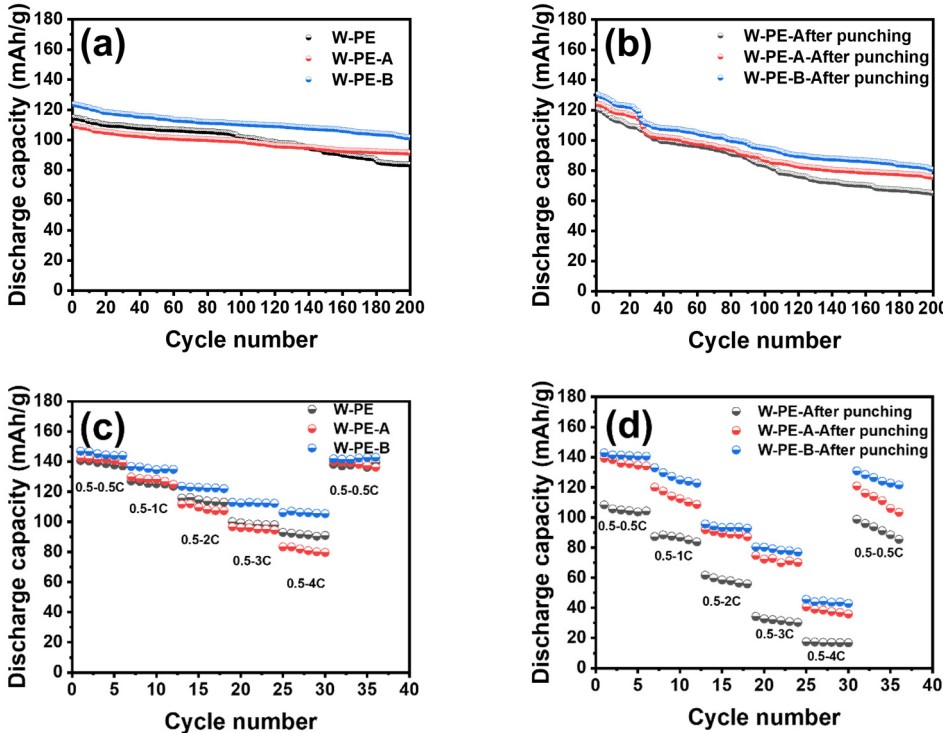

**Figure 7.** Cyclic charge/discharge performance of three separators (0.5 C charge–2 C discharge; cut-off voltage: 2.5–4.2 V) before (**a**) and after cyclic compression (**b**); C-rate performance of three separators before (**c**) and after cyclic compression under 60 °C (**d**).

Interfacial resistance has a direct impact on the C-rate performance, and Figure 7a shows that the W-PE-A separator had the highest interfacial resistance, and the cells assembled with this separator exhibited the worst C-rate performance, with the capacity decreasing more as the rate increased. In particular, the discharge capacity decreased by 41% at a rate of 4 C, while the discharge capacity of the W-PE and W-PE-B separator-assembled batteries decreased by 29% and 26%, respectively, at a rate of 4 C, proving that the contact between the separator and the electrolyte and the interfacial compatibility

greatly affects the C-rate performance of the battery. Compared with alumina particles, the choice of boehmite particles for coating improves the interface resistance, resulting in better assembled battery rate performance.

Figure 7b shows that the three separators after cyclic compression at 60 °C, when assembled into the cells, had a higher initial discharge capacity than the separators that were not cycled and compressed. This is due to the fact that the microporous structure of the separator is destroyed after compression, and the "branch traction" effect makes the pore size larger in certain areas, thus making ion transport easier, but also leads to the risk of micro-short circuits. After 200 cycles, the cyclic charge/discharge performance compared to the separator without cyclic compression was very different, with capacity retention rates of approximately 50%, 65%, and 67%, respectively.

The significant decrease in capacity retention compared to the separator without cyclic compression demonstrates that the structural damage of the microporous separator affects the cyclic charge/discharge performance of the battery. The separator coated with alumina and boehmite particles could still have a capacity higher than 80 mAh/g after 200 cycles after cyclic compression, and the charge and discharge performance was more stable at different rates.

In Figure 8(a1,b1,c1), the solid line indicates the discharge curve, and the dashed line indicates the charging curve. As can be seen from the discharge curves, as the C-rate increases, the discharge plateau of the cell becomes narrower, and at 4 C, the boehmite-coated separator presents a higher discharge capacity. There is a huge link between this and the interfacial resistance of the separator, which presents better rate performance when it has better physical contact with the electrolyte and the positive and negative electrodes and a lower interfacial resistance. According to Figure 4, it can also be seen that the W-PE-B separator has the better ability to charge and discharge at high current densities.

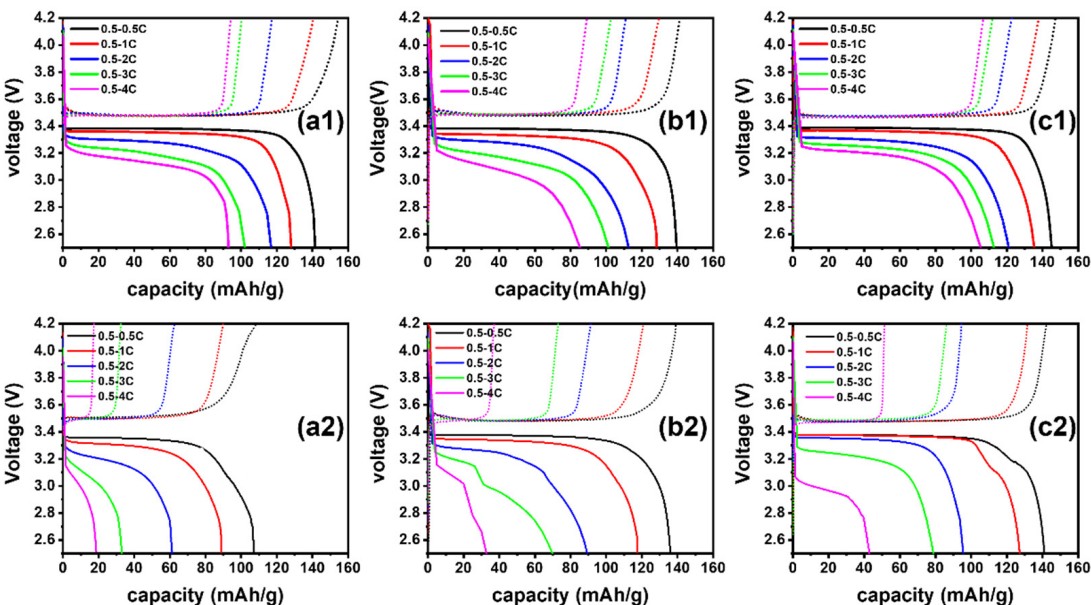

**Figure 8.** Charge and discharge curves of C-rate test of W-PE separator (**a1**), W-PE-B separator (**b1**), and W-PE-B separator (**c1**) before cyclic compression; charge and discharge curves of C-rate test of W-PE separator (**a2**), W-PE-B separator (**b2**), and W-PE-B separator (**c2**) after cyclic compression under 60 °C.

In Figure 8(a2,b2,c2), the compressed separator does not present a stable discharge plateau at a high rate. Due to the structural damage, the maximum oxidative voltage limit drops, and neither can perform a stable charge/discharge cycle at a high rate. The initial discharge voltage also drops from around 3.4 V to below 3.0 V. However, at a low rate, the assembled battery is capable of cyclic charging and discharging at the oxidative voltage

limit. Therefore, for the separator coated with inorganic particles in the liquid electrolyte environment, the separator structure is protected. In particular, the separator coated with boehmite particles can maintain the original charge and discharge performance at a rate of 0.5 C and maintain the smooth operation of the battery.

### 3.3. Microstructure and Electrochemical Performance of Three Separators before and after Cyclic Compression under Room Temperature

Figure 9 shows the microstructures of the three separators after cyclic compression under room temperature. Compared with Figure 2, there are differences in the apparent topography of the separators after cyclic compression at room temperature and 60 °C. The pore structure of the polyethylene separator is not destroyed. After the coated separators were compressed at room temperature, the wear effect of the coating particles on the pore structure in the separators was weakened. Therefore, in addition to considering the coupling of the compressive stress and solvent field in the process of battery service, temperature is also one of the indispensable factors.

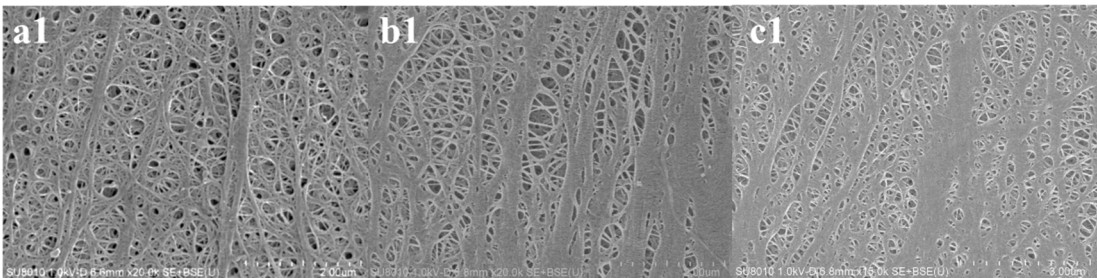

**Figure 9.** The surface SEM of W-PE (**a1**), W-PE-A (**b1**), and W-PE-B (**c1**); in the latter two samples, the surface particles after cyclic compression under room temperature were washed away.

It can be seen from Figure 10a that after assembling the battery using the separators after cyclic compression at room temperature, the interface resistance increased by more than 100 ohms compared with the original base separator, but compared with the separator after cyclic compression under 60 °C, the interface resistance was much smaller, which is due to the lack of temperature effects during the experiment. The steady-state voltage of the coated film shown in Figure 10b is above 4.5 V, indicating that its structural integrity is preserved and the liquid electrolyte is not decomposed in the operating voltage range, which ensures that the battery can be reliably cycled in the normal operating voltage range. In summary, in the process of studying the relationship between the change of the separator microstructure and the electrochemical performance during battery service, the temperature field, as an important factor, deserves our attention.

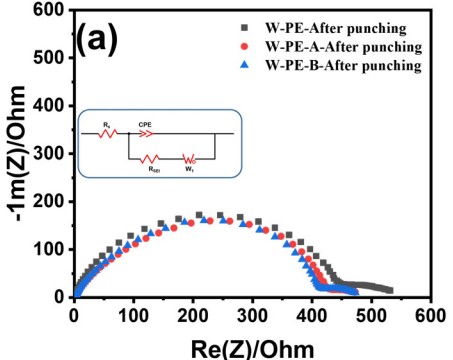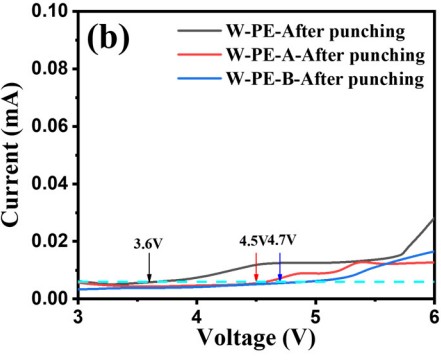

**Figure 10.** (**a**) Interfacial resistance of three separators after cyclic compression under room temperature with Li/separator–liquid electrolyte/Li. (**b**) Linear scanning voltammograms of three separators after cyclic compression under room temperature using saturated 1 M LiPF$_6$ as electrolyte.

## 4. Conclusions

The service aging process of the separator under battery operating conditions was simulated, and the relationship between the microstructural change of the separator and the ion transport and discharge capacity after assembling the battery were compared. For the aging behavior of three commercial separators under room temperature, the microporous structure was not significantly damaged, and the interfacial resistance was not greatly affected, suggesting that it is more appropriate to analyze the aging behavior of the separator under battery operating temperature conditions.

For the polyethylene separator compressed under 60 °C in an electrolyte environment, the porous skeleton was partially ruptured, and the average pore size increased, not only affecting the ion transport channel but also reducing the ion conductivity by an order of magnitude. In addition, the stable voltage of the battery was lower than the normal discharge voltage range, and the discharge capacity also dropped very seriously.

For the coated separators, the steady-state voltage remained above 4.2 V after cycling compression under 60 °C in an electrolyte environment, and the discharge capacity was only lost by about 25% after 200 cycles. Especially for the separator coated by boehmite, the electrochemical performance after cyclic compression under 60 °C was better. Hence, the choice of coating particles should be included for the separator to face the complicated compression stress and a high-temperature electrolyte environment under the working state of the battery, to ensure its long-term electrochemical performance.

**Supplementary Materials:** The following supporting information can be downloaded at: https://www.mdpi.com/article/10.3390/coatings14040419/s1, Figure S1: Linear scanning voltammograms (0–6.0 V) of three separators using saturated 1 M LiPF$_6$ as electrolyte before (left) and after cyclic compression under 60 °C (right); Figure S2: SEM photographs of the uncoated surface for W-PE-A (left) and W-PE-B (right) without cyclic compression.

**Author Contributions:** Conceptualization, Y.W.; Methodology, W.Q.; Validation, C.L.; Resources, S.W.; Writing—original draft, W.Q.; Writing—review & editing, C.L. and R.X. All authors have read and agreed to the published version of the manuscript.

**Funding:** This research received no external funding.

**Institutional Review Board Statement:** Not applicable.

**Informed Consent Statement:** Not applicable.

**Data Availability Statement:** Data are contained within the article.

**Conflicts of Interest:** The authors Shuqiu Wu and Yanjie Wang were employed by Shenzhen Senior Technology Material Co., Ltd., and remaining authors declare no conflict of interest.

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
