# Peer review of "Aging Behavior of Polyethylene and Ceramics-Coated Separators under the Simulated Lithium-Ion Battery Service Compression and Temperature Field"

_coatings, doi:10.3390/coatings14040419_

Round 1
Reviewer 1 Report
Comments and Suggestions for Authors
The authors tried to analyses the of three commercial polyethylene separators on performance in lithium-ion battery. The polyethylene separator and the alumina or boehmite coated separators were selected for simulated adding process and the relationship between the microstructure change of the separator and the ion transport and discharge capacity after assembling the battery were compared. This is interesting to the community and the work supplies some useful information. The paper can be accepted after considering the following issues/
1. I would suggest the authors to add information about how the surface particles were washed away from PE separators.
2. line 175 instead of “ceramic particles” use “oxide ceramic particles”.
3. More detailed discussion should be added about interface resistance on separators coated with Al2O3 and boehmite. On fig. 2 both particles looks the same.
Author Response
Please see the attachment:
- I would suggest the authors to add information about how the surface particles were washed away from PE separators.
Response to comment:I added the test method to the manuscript.In order to further observe the surface morphology of the coated separators after cyclic compression at 60°C, the separators were soaked in deionised water and ultrasonically cleaned.
- line 175 instead of “ceramic particles” use “oxide ceramic particles”.
Response to comment:The comments made have been revised in the manuscript.
- More detailed discussion should be added about interface resistance on separators coated with Al2O3 and boehmite. On fig. 2 both particles looks the same.
Response to comment:Aluminium and boehmite are both oxides, one with a twelve-dimensional three-dimensional structure and one with a three-dimensional three-dimensional structure, and there is a great deal of similarity in terms of morphology, but it is possible to see that the alumina particles are more similar to lumps, and that the boehmite particles are in the form of flakes.Boehmite has better compatibility with organic solvents compared to alumina powders. In addition, its overall lamellar outer body structure will improve the coating flatness, which ensures that the contact between the diaphragm coated with Boehmite particles and the lithium sheet will be better. This also verifies that the W-PE-B separator has the lowest interfacial resistance in the test results. The higher resistance of W-PE-A separator is due to the alumina blocky particle morphology that is not conducive to interfacial contact.

Reviewer 2 Report
Comments and Suggestions for Authors
In this paper, a device was set up, which could simulate the separator environment in the battery to track the influence of compression, temperature and electrolyte on the structure and electrochemical performance of separators. The commercial polyethylene separator and the alumina or boehmite coated separators were selected, and the high-temperature cyclic compression was carried out in a mixed solvent environment with a ratio of vinyl carbonate and diethyl carbonate of 1:1. Compared with that compressed for 50 cycles under room temperature, the compression at 60 °C resulted in the pore structure deterioration in the polyethylene separator. The steady-state voltage was reduced to 3.5V, and after 200 charge and discharge cycles, the capacity was reduced by more than 50%. However, for the coated separator, the presence of coating layer exhibited some protective effect, the microporous structure in the base membrane was preserved. The steady-state voltage was above 4.0V. At 4C charge and discharge rate, the coating particles were found to insert in the microporous structure, resulting in the decrease of porosity and discharge capacity. Compared with that coated with alumina particles, the interface resistance for separator coated with boehmite particles was minimally affected and the electrochemical performance after cyclic compression under 60 °C was better, exhibiting higher application ability.
The paper could be considered for publication in the Journal after the following major revisions:
1-move these sentences to introduction:
“The polyolefin separators not only encounter compressive stresses under the working state of lithium-ion batteries, but also face high temperature in the electrolyte environment. Their aging would lead to deterioration of battery performance and even battery failure. However, the relationship between the structure changes of separators and battery aging has not been clearly built up. “
2-Define in the abstract what parameters were investigated, before briefly mentioning the results of such tests. Also, reduce the very detail of the work. It is too wordy now.
3-Check the English of the whole paper.
4-employ a better combination of the keywords.
5-Introduciton should be strengthened. To modify this section the following documents can be consulted:
10.1038/s41557-018-0045-4 , doi: https://doi.org/10.1002/adfm.202303466
6-it should be “materials and experimental procedure”. And separate the two sections as well.
7-also, talk about the number tests performed for each microhardness test.
8-reference the formula that are used.
9-explain the features on figures. Some figure captions are long and tedious.
10-consult the following references in the discussion section:
doi: https://doi.org/10.1016/j.apsusc.2023.158639 , doi: 10.1002/aenm.201502588
11-in figure 6b, the two lines are almost overlapping. Any explanation for this ?
12-make the lines narrower in figure7a,b so that the lines could be differentiated.
13-saparate the images in figure 9. Currently, they all look from the same region. Explain the features on them.
14-strengthen the discussion.
15-conclusions should be more concise and not just mention the results. They are too wordy and tedious. Also made them quantitative. They should be in bullet points.
16-better describe the rationale of the work at the end of the introduction. Despite a lengthy introduction, the main rationale of the work is not clearly stated. Just pin point the main reason(s).
Comments on the Quality of English Language
3-Check the English of the whole paper.
Author Response
Please see the attachment
The paper could be considered for publication in the Journal after the following major revisions:
1-move these sentences to introduction:
“The polyolefin separators not only encounter compressive stresses under the working state of lithium-ion batteries, but also face high temperature in the electrolyte environment. Their aging would lead to deterioration of battery performance and even battery failure. However, the relationship between the structure changes of separators and battery aging has not been clearly built up. “
2-Define in the abstract what parameters were investigated, before briefly mentioning the results of such tests. Also, reduce the very detail of the work. It is too wordy now.
3-Check the English of the whole paper.
4-employ a better combination of the keywords.
5-Introduciton should be strengthened. To modify this section the following documents can be consulted:
10.1038/s41557-018-0045-4 , doi: https://doi.org/10.1002/adfm.202303466
Response to comments1-5: Based on the reviewer's comments and requests,I revised the abstract section and checked the level of English throughout the manuscript, better combining keywords, and revised the introduction.
6-it should be “materials and experimental procedure”. And separate the two sections as well.
7-also, talk about the number tests performed for each microhardness test.
8-reference the formula that are used.
9-explain the features on figures. Some figure captions are long and tedious.
10-consult the following references in the discussion section:
doi: https://doi.org/10.1016/j.apsusc.2023.158639 , doi: 10.1002/aenm.201502588
Response to comments6-10: Based on the reviewer's comments and requests,I referenced the references mentioned by the reviewer, revised the section on materials and experimental procedures and combined them more logically, added citations to formulas, and simplified numerical features.
- in figure 6b, the two lines are almost overlapping. Any explanation for this ?
Response to comment:When performing the LSV test steady state voltage, the front end is all smooth and the vertical coordinates represent currents that are all small and close together, which can be present in the test, but do not appear to be close together when the vertical coordinates are scaled up.
- make the lines narrower in figure7a,b so that the lines could be differentiated.
Response to comment:I have revised the Figure 7 in the manuscript
13-saparate the images in figure 9. Currently, they all look from the same region. Explain the features on them.
Response to comment:Both coated films were coated with wet manufactured polyethylene film as the substrate, so when the surface coating was washed off with deionised water there was not much difference in their microporous structure, which indicates that cyclic compression at room temperature does not damage the microporous structure very much, and so the electron microscope picture shows that there is not a great deal of difference in their microporous structure.
14-strengthen the discussion.
Response to comment:I have intensified the discussion in the manuscript
15-conclusions should be more concise and not just mention the results. They are too wordy and tedious. Also made them quantitative. They should be in bullet points.
Response to comment:I have made changes in the original manuscript to simplify the conclusions.
16-better describe the rationale of the work at the end of the introduction. Despite a lengthy introduction, the main rationale of the work is not clearly stated. Just pin point the main reason(s).
Response to comment: I have presented the main rationale for this work at the end of the abstract and have rewritten the abstract section.

Reviewer 3 Report
Comments and Suggestions for Authors
Recommendation: Minor revision
1. The abstract should be more informative.
2. The first letter of keywords should be capital letters.
3. The adsorption/desorption isotherm should be provided.
4. XRD patterns of the proposed sample should be provided.
5. In Figure 4, an equivalent circuit diagram should be provided.
Comments on the Quality of English LanguageMinor editing of the English language is required.
Author Response
Please see the attachment
- The abstract should be more informative.
Based on the reviewer's suggestions and requests, I have made changes to the abstract section in the manuscript.
- The first letter of keywords should be capital letters.
Based on the reviewer's suggestions and requests, I have made revisions in the manuscript.
- The adsorption/desorption isotherm should be provided.
The reviewer suggested providing adsorption/desorption isotherms, which I understand to be differential scanning calorimetry (DSC) curves. By using DSC 3 (METTLER TOLEDO) to characterize the three separators at a heating rate of 10°C min-1, it is beyond doubt that the melting temperature of the separators has increased after application. This is the same as the XRD test, which is a test that has already been done after the sample is available. However, it has little relevance to the aging behavior and electrochemical properties of the separator that is intended to be described in the manuscript, so it is not released in the manuscript.The DSC curve is shown in the figure.
- XRD patterns of the proposed sample should be provided.
Three commercial separators were supplied by Shenzhen Senior Technology Material Co. Ltd.These tests in the manuscript have been preceded by XRD testing of three samples to determine whether they are polyethylene principles and to determine the properties of the coated particles, as shown in the figure.

Reviewer 4 Report
Comments and Suggestions for Authors
The article deals with a recent interesting topic about battery research. I have some comments to give to the authors.
Similarity Analysis
The authors should paraphrase correctly the text in lines 66-69 and lines and the characterization section from lines 130-160 as a considerable percentage of similarity with other works is detected.
Units
In some parts of the text the units appear without an space after the number (example: 3.5V in line 21,24, 135, 265, 284, 365, 383) and in line 267 appears with a space (2.5 V). According to NIST, a space should be included between the numerical value and the unit symbol. This mistake appears repeatedly throughout the text.
The authors should carefully check the significant digits authors express. For example in line 279 they write 4.0 V and 4V in the same line. The same issue occurs in line 338.
Concepts
In line 27-28 the authors affirm that the electrochemical performance after cyclic compression under 60 °C was better, exhibiting higher application ability.
The authors should be more specific with the terms “better” and “application ability”. I also suggest being more specific with the terms “germination” in line 39 and “excitation” in line 44.
Please also check if “absorption rate” in line 291 is the correct term.
Materials and Methods
In section 2.1 there are some words in bold that need to be changed
The authors should provide more details about the determination of pore size distribution, as it is not very clear which method was used to measure that property.
SEM
From SEM micrographs I consider it surprising that both ceramic coatings of W-PE-A and W-PE-B completely disappear after treatment both at 25 and 60 °C. I think the authors should provide more information and explanation about this fact.
Figures
I recommend relocating figures 5, 6, 7, 8 and 9 as they are located before they are mentioned in the text.
Figures 2 and 9 should be adjusted to the margins of the template.
Figure 7 should be in one entire page.
In Figures 2 and 9 the legend of both figures say: “surface particles were washed away”. Perhaps I did not understand clearly the reason that comment is in the legend, but I understand it is about sample preparation before SEM analysis. If that’s the case my suggestion is to include it in the characterization section or indicate it in the result analysis instead of being part of the legend of the figures.
Ionic conductivity
In lines 213 and 214 the authors report the values of ionic conductivity before cyclic compression. I am quite surprised tha the measured values are very close to the ionic conductivity of pure liquid electrolytes. My suggestion is to relate with conductivity values found in literature of other separators and compare them. Moreover, I want to know if the fact that the authors remove contact and wire resistance has a considerable impact on the value of ionic conductivity.
Section 3.3 Efect of temperature
I would like to know the reason authors did not perform analysis of cell performance at 25 °C, is because of ionic conductivity? I think the authors should complement this information.
The authors mention that the interface resistance is much smaller due to the effect of temperature. My suggestion is to include the values to facilitate comparison for the reader.
The effect on the temperature is not summarized in the conclusions.
References
Change the reference number [20] as it appears superindexed.
Comments on the Quality of English LanguageIn line 286, I think it should say “for the three separators” instead of “with three separator”
Check if line 352 is “separator” or “separators” as it is not very clear which sample the authors are describing.
Change “Contact Angle (°)” instead of “The contact angle (º)”
Author Response
Please see the attachment
Comments and Suggestions for Authors
The article deals with a recent interesting topic about battery research. I have some comments to give to the authors.
Similarity Analysis
The authors should paraphrase correctly the text in lines 66-69 and lines and the characterization section from lines 130-160 as a considerable percentage of similarity with other works is detected.
Response to comment:Based on the reviewer's suggestions and requests, I have rewritten the mentioned part in the manuscript.
Units
In some parts of the text the units appear without an space after the number (example: 3.5V in line 21,24, 135, 265, 284, 365, 383) and in line 267 appears with a space (2.5 V). According to NIST, a space should be included between the numerical value and the unit symbol. This mistake appears repeatedly throughout the text.
The authors should carefully check the significant digits authors express. For example in line 279 they write 4.0 V and 4V in the same line. The same issue occurs in line 338.
Response to comments:Based on the reviewer's suggestions and requests, I have checked the significant figures carefully and modified it in the manuscript.
Concepts
In line 27-28 the authors affirm that the electrochemical performance after cyclic compression under 60 °C was better, exhibiting higher application ability.
The authors should be more specific with the terms “better” and “application ability”. I also suggest being more specific with the terms “germination” in line 39 and “excitation” in line 44.
Please also check if “absorption rate” in line 291 is the correct term.
Response to comments:
The germination is changed to the “generation” and the terminology is used more specifically in the process of expression.
I've double-checked the words “absorption rate” and changed it to the aspiration rate .
Materials and Methods
In section 2.1 there are some words in bold that need to be changed
The authors should provide more details about the determination of pore size distribution, as it is not very clear which method was used to measure that property.
Response to comment: I have mentioned the test method for pore size distribution in the test method and provided relevant details on the pore size distribution.
SEM
From SEM micrographs I consider it surprising that both ceramic coatings of W-PE-A and W-PE-B completely disappear after treatment both at 25 and 60 °C. I think the authors should provide more information and explanation about this fact.
Response to comment:In fact, there was a slight residue in the overall morphology after treatment, but since the magnification was adjusted to see the changes in the micropore structure, the particle residue was rarely visible, and the SEM image with no particle residue was specially selected.
Figures
I recommend relocating figures 5, 6, 7, 8 and 9 as they are located before they are mentioned in the text.
Figures 2 and 9 should be adjusted to the margins of the template.
Figure 7 should be in one entire page.
In Figures 2 and 9 the legend of both figures say: “surface particles were washed away”. Perhaps I did not understand clearly the reason that comment is in the legend, but I understand it is about sample preparation before SEM analysis. If that’s the case my suggestion is to include it in the characterization section or indicate it in the result analysis instead of being part of the legend of the figures.
Response to comments:I have adjusted the position of the mentioned images, margins, etc., according to the comments and requests of the reviewers.
Ionic conductivity
In lines 213 and 214 the authors report the values of ionic conductivity before cyclic compression. I am quite surprised tha the measured values are very close to the ionic conductivity of pure liquid electrolytes. My suggestion is to relate with conductivity values found in literature of other separators and compare them. Moreover, I want to know if the fact that the authors remove contact and wire resistance has a considerable impact on the value of ionic conductivity.
Response to comments:The ionic conductivity of the pure liquid electrolyte is in the order of magnitude of 10-3~ 10-2, and the ionic transport of the diaphragm itself is also through the active ions in the liquid electrolyte using the solvent as a carrier in its internal channels, so its ionic conductivity is very close.I will also make changes in the manuscript to compare it to the ionic conductivity of other diaphragms.
The removal of contact and wire resistances is intended to further the accurate calculation of ionic conductivity. Because of the test setup, these resistances do exist and cannot be ignored during the test. The removal of the contact and wire resistances does not have a significant effect on the calculated ionic conductivity values because the same test setup is used for each of the resistances and these resistances are subtracted from the calculations and are small.
Section 3.3 Efect of temperature
I would like to know the reason authors did not perform analysis of cell performance at 25 °C, is because of ionic conductivity? I think the authors should complement this information.
The authors mention that the interface resistance is much smaller due to the effect of temperature. My suggestion is to include the values to facilitate comparison for the reader.
The effect on the temperature is not summarized in the conclusions.
Response to comments:The ionic conductivity of the diaphragm after cyclic compression at 25 degrees Celsius was not significantly different compared to that without cyclic compression and the microporous structure was not destroyed, so it was not analysed in the electrochemical properties specifically.
Based on the reviewers' suggestions, I will make changes in the manuscript.
In the conclusion, I will summarise the effect of temperature .
References
Change the reference number [20] as it appears superindexed.
Response to comment:I've changed the reference.
Comments on the Quality of English Language
In line 286, I think it should say “for the three separators” instead of “with three separator”
Check if line 352 is “separator” or “separators” as it is not very clear which sample the authors are describing.
Change “Contact Angle (°)” instead of “The contact angle (º)”
Response to comments:I checked the quality of the language and adjusted the words mentioned.

Reviewer 5 Report
Comments and Suggestions for Authors
The authors present a timely study into the combined effects of compressive stress and elevated temperatures on lithium-ion battery separators. The work is of interest to a general readership, and includes some context of relevance to coatings in that it considers coated separators.
The introduction is well written and concise. The experimental section is lacking some critical details (see below). The results and discussion section is adequately organised and relatively clear. The figures are for the most part clear and understandable. The conclusions section is impactful and concise. The references cited in the article are relevant, up to date, and represent the current literature well.
Overall, the article is of interest, but there are several unclear points, particularly in the experimental section and early on in the results/discussion sections that render detailed understanding of the later data (electrochemical measurements etc) difficult. I would like to ask the authors to clarify according to the following points, and then I would be delighted to further consider the manuscript in its entirety.
1. The electrolyte being used is little unclear. In the abstract (line 18) and also on line 109, "vinyl carbonate" is mentioned, but at many other points in the manuscript, EC/DEC is mentioned. It is also rather unclear as to which experiments use 1M LiPF6 in mixed solvent, and which use just mixed solvent (i.e. no salt). e.g. line 118, line 141, and the reasons for this.
2. In the caption of Fig 2, is mentioned that the surface particles of the cyclic compression-treated separators with coating were washed away before SEM images were recorded. It is not clear if this washing process was also carried out before other experiments with these separators e.g. pore size distribution, electrochemical tests etc. I would like to ask the authors to make this clear. The presence or absence of the residual particles from the surface coating will make a great difference to the results, and so it is difficult to make further comment on the subsequent data without this being clarified.
3. In Fig 2, the SEM images for W-PE-A and W-PE-B before cyclic compression, and also after cyclic compression (and removal of particles) are shown. Is it possible to see these separators with the coating removed, but without cycic compression? This would clarify to what extent the observed morphology in b2 and b3 are due to the cyclic compression, rather than pre-existing due to the coating process. I also noticed that b2 and b3 are not really described in the text. The scale bar for b3 is also missing.
4. Pore size distribution is mentioned in the text around line 192 (and Fig. 3), but I couldn't find any experimental description (in the Experimental section) for how these data were obtained. Please describe in full. If possible, can data for W-PE-A and W-PE-B before cyclic compression (punching) be included?
Additionally there are some omissions/corrections
5. Line 107: Is the LFP cathode purchased as-is? What is the binder, conductive carbon used? Is the current collector Al, or carbon-coated Al? Is the electrode calendared (rolled/pressed)? If so, what are the conditions?
6. Line 107: What are "lithium flakes"? Is it lithium sheet?
7. Line 114: what material is the hemispherical indenter? PTFE?
8. Line 130: What is the contact angle measurement temperature? Were the separators soaked by full immersion in the electrolyte, or just placed on the surface of the liquid? If on the surface, for the coated separators was the coating on the upper (not in contact with liquid) or lower (in contact with liquid) face?
9. Equation 1 and Table 1: The % electrolyte uptake reflects the change in mass, but to fully understand, the reader should have an idea of the initial mass. Please include the average wdry values in the table.
10. Line 148-170: The electrolyte is not mentioned (what electrolyte and what volume). The reader should also be informed of the diameter of the electrodes and separator.
11. Line 150: What is the voltage perturbation used for the EIS?
12. Line 159: Why is the LSV starting at 0 V? The assembled cell with SS working electrode and Li counter/reference will typically have an open circuit voltage of very roughly 2 V. Stepping this down to 0 V will result in unknown reductive processes at the working electrode, and the effect of these reductive processes (leading to products in solution and/or on the electrode surface) on the subsequent positive voltage scan is unknown. A more typical protocol for this type of LSV would be to start directly at the open circuit voltage, or initially step to a voltage slightly higher than the open circuit voltage. The voltage range of interest is likely 3 to 6 V, so it is not clear to me why it is necessary to start from 0 V, but if the authors have a different idea I would be happy to hear it.
13. Figs 5 and 10: ideally, EIS data should be presented in square (not rectangular) graphs, with the same numerical range for x-axis and y-axis (to easily visualise any deviation from perfect semicircle)
14. Line 268: "proved" Please don't use "proof"/"prove"/"proved", these are absolute expressions which should not be used in the context used here. The authors intended meaning would be better represented by "This supports the proposal that the coated separator...", or "This demonstrates that the coated separator..." or similar phrasing.
Author Response
Please see the attachment
- The electrolyte being used is little unclear. In the abstract (line 18) and also on line 109, "vinyl carbonate" is mentioned, but at many other points in the manuscript, EC/DEC is mentioned. It is also rather unclear as to which experiments use 1M LiPF6 in mixed solvent, and which use just mixed solvent (i.e. no salt). e.g. line 118, line 141, and the reasons for this.
Response to comments:Vinyl carbonate is abbreviated as EC and diethyl carbonate is abbreviated as DEC, which have been standardised as EC and DEC in the manuscript.The liquid electrolyte of lithium-ion battery includes lithium salt and solvent, the lithium salt is LiPF6, which plays the role of providing active lithium ions after dissolved in the solvent, in the cyclic compression process, taking into account the environmental safety, so only the synergistic effect of the solvent field is taken into account, using a mixed solvent of EC/DEC, and the other tests are all using 1M LiPF6.
- In the caption of Fig 2, is mentioned that the surface particles of the cyclic compression-treated separators with coating were washed away before SEM images were recorded. It is not clear if this washing process was also carried out before other experiments with these separators e.g. pore size distribution, electrochemical tests etc. I would like to ask the authors to make this clear. The presence or absence of the residual particles from the surface coating will make a great difference to the results, and so it is difficult to make further comment on the subsequent data without this being clarified.
Response to comments:No cleaning of the particles was carried out for the other experiments, except that the samples were cleaned before taking the SEM images. The aim of this experiment was to find out whether the ageing behaviour of the coated modified separator would be better in terms of retaining certain electrochemical properties compared to the uncoated pure polyethylene separator. Secondly, the unavoidable plugging of the holes by a few particles was analysed to see if it would directly cause the battery to fail, thus determining that coating the separator with oxide particles would not only help to increase the discharge capacity, but also slow down the ageing behaviour and increase the number of charge/discharge cycles to a certain extent.
- In Fig 2, the SEM images for W-PE-A and W-PE-B before cyclic compression, and also after cyclic compression (and removal of particles) are shown. Is it possible to see these separators with the coating removed, but without cycic compression? This would clarify to what extent the observed morphology in b2 and b3 are due to the cyclic compression, rather than pre-existing due to the coating process. I also noticed that b2 and b3 are not really described in the text. The scale bar for b3 is also missing.
Response to comments:Because the substrate of the coated separator is W-PE film, the microporous structure of the W-PE film is shown after removing the coating, which is similar to the SEM image of the W-PE film, so it is not shown on the SEM image. a2,a3 shows the surface topography of the coated separator without cyclic compression, and the particle morphology is obvious. I will enhance the description of b2,b3 in the manuscript.
- Pore size distribution is mentioned in the text around line 192 (and Fig. 3), but I couldn't find any experimental description (in the Experimental section) for how these data were obtained. Please describe in full. If possible, can data for W-PE-A and W-PE-B before cyclic compression (punching) be included?
Additionally there are some omissions/corrections
Response to comments:I have added a specific description in the experimental section and other reviewers have mentioned the same issue. In addition, to address the issue of the data of W-PE-A and W-PE-B before cycling cycle, since the gap of the coating particles will be larger than the pore size of the microporous membrane when the computer analysis is performed by this test method, it will lead to an inaccurate and not very meaningful measurement of the pore size distribution. On the other hand, since both coated films use the same W-PE film as the substrate, the measured pore size distribution after cleaning the coating particles is the same as that of the W-PE film, so it was not tested.
- Line 107: Is the LFP cathode purchased as-is? What is the binder, conductive carbon used? Is the current collector Al, or carbon-coated Al? Is the electrode calendared (rolled/pressed)? If so, what are the conditions?
lithium iron phosphate cathode (ratio of active substance: 91.5%)is purchased from Guangdong Canrd New Energy Technology Co. Ltd., China.Specific information has been added to the material section with specific instructions.
- Line 107: What are "lithium flakes"? Is it lithium sheet?
Response to comment:I've made changes in the manuscript.
- Line 114: what material is the hemispherical indenter? PTFE?
Response to comment:The hemispherical indenter in customised stainless steel.
- Line 130: What is the contact angle measurement temperature? Were the separators soaked by full immersion in the electrolyte, or just placed on the surface of the liquid? If on the surface, for the coated separators was the coating on the upper (not in contact with liquid) or lower (in contact with liquid) face?
Response to comments:The contact angle measurement is a test performed at room temperature. The angular measurement value of the contact angle is measured by dropping the electrolyte onto the surface of the separator using a syringe for electrolyte titration, and by capturing the angle formed by the electrolyte drop onto the surface of the separator, the affinity with the electrolyte is judged.
- Equation 1 and Table 1: The % electrolyte uptake reflects the change in mass, but to fully understand, the reader should have an idea of the initial mass. Please include the average wdryvalues in the table.
Response to comment:The initial mass is calculated by cutting separators of the same size and calculating the absorption rate after several tests, and then averaging the values to be more convincing, rather than calculating by averaging the masses.
- Line 148-170: The electrolyte is not mentioned (what electrolyte and what volume). The reader should also be informed of the diameter of the electrodes and separator.
Response to comment:Among them, due to the selection of CR2032 battery shell assembly, so the diameter of the lithium sheet is 16mm, the diameter of the separator is 20mm, the diameter of the steel sheet is 18mm,the liquid electrolyte (1M LiPF6 in EC/DEC, volume ratio 1:1).
- Line 150: What is the voltage perturbation used for the EIS?
Response to comment:Disturbance voltage amplitude is 5mV.
- Line 159: Why is the LSV starting at 0 V? The assembled cell with SS working electrode and Li counter/reference will typically have an open circuit voltage of very roughly 2 V. Stepping this down to 0 V will result in unknown reductive processes at the working electrode, and the effect of these reductive processes (leading to products in solution and/or on the electrode surface) on the subsequent positive voltage scan is unknown. A more typical protocol for this type of LSV would be to start directly at the open circuit voltage, or initially step to a voltage slightly higher than the open circuit voltage. The voltage range of interest is likely 3 to 6 V, so it is not clear to me why it is necessary to start from 0 V, but if the authors have a different idea I would be happy to hear it.
Response to comment:I didn't carefully consider the reduction process of the working electrode, and when I was looking up test methods earlier there was literature on tests done in the voltage interval of 0-6V, with the intention of seeing a smoother LSV curve .
- Figs 5 and 10: ideally, EIS data should be presented in square (not rectangular) graphs, with the same numerical range for x-axis and y-axis (to easily visualise any deviation from perfect semicircle)
Response to comment:I'll adjust the horizontal and vertical coordinates in the EIS plot to make it easier to see any deviations from a perfect semicircle
- Line 268: "proved" Please don't use "proof"/"prove"/"proved", these are absolute expressions which should not be used in the context used here. The authors intended meaning would be better represented by "This supports the proposal that the coated separator...", or "This demonstrates that the coated separator..." or similar phrasing.
Response to comments:I have checked the phrasing in the manuscript to tighten it up.

Round 2
Reviewer 2 Report
Comments and Suggestions for Authors
My previous comments were not fully implemented. The following major revisions need to be applied fully before I make a final judgment regarding the suitability of the paper for publication in the Journal:
In this paper, a device was set up, which could simulate the separator environment in the battery to track the influence of compression, temperature and electrolyte on the structure and electrochemical performance of separators. The commercial polyethylene separator and the alumina or boehmite coated separators were selected, and the high-temperature cyclic compression was carried out in a mixed solvent environment with a ratio of vinyl carbonate and diethyl carbonate of 1:1. Compared with that compressed for 50 cycles under room temperature, the compression at 60 °C resulted in the pore structure deterioration in the polyethylene separator. The steady-state voltage was reduced to 3.5V, and after 200 charge and discharge cycles, the capacity was reduced by more than 50%. However, for the coated separator, the presence of coating layer exhibited some protective effect, the microporous structure in the base membrane was preserved. The steady-state voltage was above 4.0V. At 4C charge and discharge rate, the coating particles were found to insert in the microporous structure, resulting in the decrease of porosity and discharge capacity. Compared with that coated with alumina particles, the interface resistance for separator coated with boehmite particles was minimally affected and the electrochemical performance after cyclic compression under 60 °C was better, exhibiting higher application ability.
1-move these sentences to introduction:
“The polyolefin separators not only encounter compressive stresses under the working state of lithium-ion batteries, but also face high temperature in the electrolyte environment. Their aging would lead to deterioration of battery performance and even battery failure. However, the relationship between the structure changes of separators and battery aging has not been clearly built up. “
2-Define in the abstract what parameters were investigated, before briefly mentioning the results of such tests. Also, reduce the very detail of the work. It is too wordy now.
3-Check the English of the whole paper.
4-employ a better combination of the keywords.
5-Introduciton should be strengthened. To modify this section the following documents can be consulted:
10.1038/s41557-018-0045-4 , doi: https://doi.org/10.1002/adfm.202303466
6-it should be “materials and experimental procedure”. And separate the two sections as well.
7-also, talk about the number tests performed for each microhardness test.
8-reference the formula that are used.
9-explain the features on figures. Some figure captions are long and tedious.
10-consult the following references in the discussion section:
doi: https://doi.org/10.1016/j.apsusc.2023.158639 , doi: 10.1002/aenm.201502588
11-in figure 6b, the two lines are almost overlapping. Any explanation for this ?
12-make the lines narrower in figure7a,b so that the lines could be differentiated.
13-saparate the images in figure 9. Currently, they all look from the same region. Explain the features on them.
14-strengthen the discussion.
15-conclusions should be more concise and not just mention the results. They are too wordy and tedious. Also made them quantitative. They should be in bullet points.
16-better describe the rationale of the work at the end of the introduction. Despite a lengthy introduction, the main rationale of the work is not clearly stated. Just pin point the main reason(s).
Comments on the Quality of English Language
Check the English of the whole paper. The sentences are too long and confusing.
Author Response
Dear Editor and Reviewers,
We would like to thank you for giving us a chance to revise (“Aging behavior of polyethylene and ceramics coated separator under the simulated lithium-ion battery service compression and temperature field”), and also thank the reviewers for giving us constructive suggestions which would help to improve the quality of the manuscript.
We have carefully checked the manuscript and modified it according to the comments. In the main text, changes have been highlighted. The point-to-point responses to the comments are as follows:
In this paper, a device was set up, which could simulate the separator environment in the battery to track the influence of compression, temperature and electrolyte on the structure and electrochemical performance of separators. The commercial polyethylene separator and the alumina or boehmite coated separators were selected, and the high-temperature cyclic compression was carried out in a mixed solvent environment with a ratio of vinyl carbonate and diethyl carbonate of 1:1. Compared with that compressed for 50 cycles under room temperature, the compression at 60 °C resulted in the pore structure deterioration in the polyethylene separator. The steady-state voltage was reduced to 3.5V, and after 200 charge and discharge cycles, the capacity was reduced by more than 50%. However, for the coated separator, the presence of coating layer exhibited some protective effect, the microporous structure in the base membrane was preserved. The steady-state voltage was above 4.0V. At 4C charge and discharge rate, the coating particles were found to insert in the microporous structure, resulting in the decrease of porosity and discharge capacity. Compared with that coated with alumina particles, the interface resistance for separator coated with boehmite particles was minimally affected and the electrochemical performance after cyclic compression under 60 °C was better, exhibiting higher application ability.
1-move these sentences to introduction:
“The polyolefin separators not only encounter compressive stresses under the working state of lithium-ion batteries, but also face high temperature in the electrolyte environment. Their aging would lead to deterioration of battery performance and even battery failure. However, the relationship between the structure changes of separators and battery aging has not been clearly built up.
Author response to comment 1:We have adjusted what the reviewer mentioned above and moved those sentences to the introduction.
- Define in the abstract what parameters were investigated, before briefly mentioning the results of such tests. Also, reduce the very detail of the work. It is too wordy now.
Author response to comment 2:The abstract section has been revised based on the reviewers' comments.
- Check the English of the whole paper.
Author response to comment 3:We have reread the entire manuscript carefully and made changes in the manuscript.
- employ a better combination of the keywords.
Author response to comment 4:We have employed the combinations of the keywords in the manuscript.
- Introduciton should be strengthened.
Author response to comment 5:We have enhanced the introduction in the manuscript.
To modify this section the following documents can be consulted:
10.1038/s41557-018-0045-4 , doi: https://doi.org/10.1002/adfm.202303466
- it should be “materials and experimental procedure”. And separate the two sections as well.
Author response to comment 6:We have separated the two sections as requested by the reviewer.
- also, talk about the number tests performed for each microhardness test.
Author response to comment 7:The experiment did not mention the microhardness test, the pore size distribution test, with a statistical number of 200 for each type of separator.
8-reference the formula that are used.
Author response to comment 8:We have added the references.
9-explain the features on figures. Some figure captions are long and tedious.
Author response to comment 9:We have explained and revised it in the manuscript.
10-consult the following references in the discussion section:
doi: https://doi.org/10.1016/j.apsusc.2023.158639 , doi: 10.1002/aenm.201502588
Author response to comment 10: We have revised it in the manuscript.
- in figure 6b, the two lines are almost overlapping. Any explanation for this ?
Author response to comment 11:The two lines almost overlap in the 3-4V voltage interval, mainly due to the large range of values in the vertical coordinate, but their raw values are different. The complete 0-6V LSV curve is shown in the above Fig. We are concerned with the oxidation process from 3-6V, so we have omitted the 0-3V data from the plotted graph.The values are close to each other, which means that their oxidation currents are close to each other at this voltage.
- make the lines narrower in figure7a,b so that the lines could be differentiated.
Author response to comment 12:We have narrowed the lines in Figure 7a and b.
- saparate the images in figure 9. Currently, they all look from the same region. Explain the features on them.
Author response to comment 13:Both coated separators were coated with wet manufactured polyethylene as the substrate, so when the surface coating was washed off with deionised water there was not much difference in their microporous structure, which indicates that cyclic compression at room temperature does not damage the microporous structure very much, and so the electron microscope picture shows that there is no apparent difference in their microporous structure.
14-strengthen the discussion.
Author response to comment 14:We have intensified the discussion in the manuscript.
- conclusions should be more concise and not just mention the results. They are too wordy and tedious. Also made them quantitative. They should be in bullet points.
Response to comment:We have made changes in the original manuscript to simplify the conclusions.
- better describe the rationale of the work at the end of the introduction. Despite a lengthy introduction, the main rationale of the work is not clearly stated. Just pin point the main reason(s).
Response to comment: We have presented the main rationale for this work and have rewritten the abstract section.
Comments on the Quality of English Language
Check the English of the whole paper. The sentences are too long and confusing.
Thanks again for your helpful and careful suggestion for my manuscript. I am looking forward to your future suggestion for my manuscript.
Sincerely yours,
Caihong Lei
School of Materials and Energy
Guangdong University of Technology
Guangzhou China 510006

Reviewer 5 Report
Comments and Suggestions for Authors
Thank you to the authors for the responses. I've provided some further comments to each response (1-14) below. In addition, following the clarifications about the analyses provided by the authors, I was able to understand the later part of the paper (porosity and electrochemical measurements) and have made some further (new) comments (15-20) related to those. I would be delighted to see the authors' further responses.
-----
1. The electrolyte being used is little unclear. In the abstract (line 18) and also on line 109, "vinyl carbonate" is mentioned, but at many other points in the manuscript, EC/DEC is mentioned. It is also rather unclear as to which experiments use 1M LiPF6 in mixed solvent, and which use just mixed solvent (i.e. no salt). e.g. line 118, line 141, and the reasons for this.
Author response to comment 1:Vinyl carbonate is abbreviated as EC and diethyl carbonate is abbreviated as DEC, which have been standardised as EC and DEC in the manuscript.The liquid electrolyte of lithium-ion battery includes lithium salt and solvent, the lithium salt is LiPF6, which plays the role of providing active lithium ions after dissolved in the solvent, in the cyclic compression process, taking into account the environmental safety, so only the synergistic effect of the solvent field is taken into account, using a mixed solvent of EC/DEC, and the other tests are all using 1M LiPF6.
Reviewer response for comment 1: I checked the manuscript and noted that there is no longer any mention of "vinyl carbonate" (which might be easily confused with vinylene carbonate, not the same as ethylene carbonate). The changes for comment 1 are appropriate and complete. Thank you.
-----
2. In the caption of Fig 2, is mentioned that the surface particles of the cyclic compression-treated separators with coating were washed away before SEM images were recorded. It is not clear if this washing process was also carried out before other experiments with these separators e.g. pore size distribution, electrochemical tests etc. I would like to ask the authors to make this clear. The presence or absence of the residual particles from the surface coating will make a great difference to the results, and so it is difficult to make further comment on the subsequent data without this being clarified.
Author response to comment 2:No cleaning of the particles was carried out for the other experiments, except that the samples were cleaned before taking the SEM images. The aim of this experiment was to find out whether the ageing behaviour of the coated modified separator would be better in terms of retaining certain electrochemical properties compared to the uncoated pure polyethylene separator. Secondly, the unavoidable plugging of the holes by a few particles was analysed to see if it would directly cause the battery to fail, thus determining that coating the separator with oxide particles would not only help to increase the discharge capacity, but also slow down the ageing behaviour and increase the number of charge/discharge cycles to a certain extent.
*Reviewer response for comment 2: Thank you. For the readers to understand better, please add the following after "... under vacuum." on line 124.
"For the coated samples, W-PE-A and W-PE-B, after cyclic compression the surface particles were washed away prior to SEM measurement. For all other measurements after the cyclic compression, this washing was not performed."
-----
3. In Fig 2, the SEM images for W-PE-A and W-PE-B before cyclic compression, and also after cyclic compression (and removal of particles) are shown. Is it possible to see these separators with the coating removed, but without cycic compression? This would clarify to what extent the observed morphology in b2 and b3 are due to the cyclic compression, rather than pre-existing due to the coating process. I also noticed that b2 and b3 are not really described in the text. The scale bar for b3 is also missing.
Author response to comment 3:Because the substrate of the coated separator is W-PE film, the microporous structure of the W-PE film is shown after removing the coating, which is similar to the SEM image of the W-PE film, so it is not shown on the SEM image. a2,a3 shows the surface topography of the coated separator without cyclic compression, and the particle morphology is obvious. I will enhance the description of b2,b3 in the manuscript.
*Reviewer response for comment 3: Thank you. Sorry if I was not clear - I am wondering if the coating process itself has some effect on the surface morphology of the film. I think it is necessary to understand this before ascribing the morphology observed in (Fig 2) b2 and b3 to being solely due to the cyclic compression. Is it possible to show the readers some SEM images of the pristine coated separators, with coating removed? The scale bar for (Fig 2) b3 is also missing, please include a scale bar for b3.
-----
4. Pore size distribution is mentioned in the text around line 192 (and Fig. 3), but I couldn't find any experimental description (in the Experimental section) for how these data were obtained. Please describe in full. If possible, can data for W-PE-A and W-PE-B before cyclic compression (punching) be included?
Author response to comment 4:I have added a specific description in the experimental section and other reviewers have mentioned the same issue. In addition, to address the issue of the data of W-PE-A and W-PE-B before cycling cycle, since the gap of the coating particles will be larger than the pore size of the microporous membrane when the computer analysis is performed by this test method, it will lead to an inaccurate and not very meaningful measurement of the pore size distribution. On the other hand, since both coated films use the same W-PE film as the substrate, the measured pore size distribution after cleaning the coating particles is the same as that of the W-PE film, so it was not tested.
*Reviewer response for comment 4: On the first point, I'm sorry, I could not find the description in the Experimental section. Please add the full experimental description for the pore size distribution measurement to the Experimental section. On the second point, I understand the authors comment, but I am still wondering whether the coating process itself causes some change in the pore structure (see also my response to comment 3 above). If possible, it is better to demonstrate to the readers, with data, the authors suggestion (in their response above) that "the measured pore size distribution after cleaning the coating particles is the same as that of the W-PE film".
-----
5. Line 107: Is the LFP cathode purchased as-is? What is the binder, conductive carbon used? Is the current collector Al, or carbon-coated Al? Is the electrode calendared (rolled/pressed)? If so, what are the conditions?
Author response to reviewer comment 5: lithium iron phosphate cathode (ratio of active substance: 91.5%)is purchased from Guangdong Canrd New Energy Technology Co. Ltd., China.Specific information has been added to the material section with specific instructions.
*Reviewer response for comment 5: Thank you. I assume the % is wt%. If so, please amend "91.5%" to "91.5wt%" in line 102.
-----
6. Line 107: What are "lithium flakes"? Is it lithium sheet?
Author response to reviewer comment 6: :I've made changes in the manuscript.
*Reviewer response for comment 6: Thank you. I found the appropriate amendment in line 103.
-----
7. Line 114: what material is the hemispherical indenter? PTFE?
Author response to reviewer comment 7: The hemispherical indenter in customised stainless steel.
*Reviewer response for comment 7: Thank you, but please inform the readers. Please add "(customised stainless steel)" after "indenter" in line 109.
-----
8. Line 130: What is the contact angle measurement temperature? Were the separators soaked by full immersion in the electrolyte, or just placed on the surface of the liquid? If on the surface, for the coated separators was the coating on the upper (not in contact with liquid) or lower (in contact with liquid) face?
Author response to reviewer comment 8:The contact angle measurement is a test performed at room temperature. The angular measurement value of the contact angle is measured by dropping the electrolyte onto the surface of the separator using a syringe for electrolyte titration, and by capturing the angle formed by the electrolyte drop onto the surface of the separator, the affinity with the electrolyte is judged.
*Reviewer response for comment 8: Thank you, but please inform the readers. Please add "at room temperature" after "tested" in line 125.
-----
9. Equation 1 and Table 1: The % electrolyte uptake reflects the change in mass, but to fully understand, the reader should have an idea of the initial mass. Please include the average wdryvalues in the table.
Author response to reviewer comment 9: The initial mass is calculated by cutting separators of the same size and calculating the absorption rate after several tests, and then averaging the values to be more convincing, rather than calculating by averaging the masses.
*Reviewer response for comment 9: Thank you for your response. If I'd understood correctly, for each piece of separator, you weigh to obtain a wdry value, then perform immersion, then weight to obtain wwet. You then calculate Electrolyte uptake (%) according to equation 1. You perform this process at least 10 times, and take an average of the Electrolyte uptake (%) values (and so you do not calculate the average wdry or wwet values). Did I understand correctly? It is not so clear from the experimental description, could you please amend. Were the diameter of separator pieces constant? (If so, please state the diameter). In that case, please also provide the typical range of values for wdry for each of W-PE, W-PE-A and W-PE-B.
By the way, there is a typo in the text, "mersed in" has slipped down into line 142, but actually belongs in line 141 ("...was imthe liquid..." should be "was immersed in the liquid")
-----
10. Line 148-170: The electrolyte is not mentioned (what electrolyte and what volume). The reader should also be informed of the diameter of the electrodes and separator.
Author response to reviewer comment 10: Among them, due to the selection of CR2032 battery shell assembly, so the diameter of the lithium sheet is 16mm, the diameter of the separator is 20mm, the diameter of the steel sheet is 18mm,the liquid electrolyte (1M LiPF6 in EC/DEC, volume ratio 1:1).
*Reviewer response for comment 10: Thank you, I found the appropriate amendment for the electrochemical measurement description in lines 157-160. Please see also my response for comment 9 above (regarding diameter for separator used for immersion test).
-----
11. Line 150: What is the voltage perturbation used for the EIS?
Author response to reviewer comment 11: Disturbance voltage amplitude is 5mV.
*Reviewer response for comment 10: Thank you, I found the appropriate amendment in the electrochemical measurement description in line 154.
-----
12. Line 159: Why is the LSV starting at 0 V? The assembled cell with SS working electrode and Li counter/reference will typically have an open circuit voltage of very roughly 2 V. Stepping this down to 0 V will result in unknown reductive processes at the working electrode, and the effect of these reductive processes (leading to products in solution and/or on the electrode surface) on the subsequent positive voltage scan is unknown. A more typical protocol for this type of LSV would be to start directly at the open circuit voltage, or initially step to a voltage slightly higher than the open circuit voltage. The voltage range of interest is likely 3 to 6 V, so it is not clear to me why it is necessary to start from 0 V, but if the authors have a different idea I would be happy to hear it.
Author response to reviewer comment 12: I didn't carefully consider the reduction process of the working electrode, and when I was looking up test methods earlier there was literature on tests done in the voltage interval of 0-6V, with the intention of seeing a smoother LSV curve .
*Reviewer response for comment 12: Thank you. However, I note that the experimental description now reads "voltage range of 2.0 to 6.0 V". Since the data shown in Fig 6 seem to be exactly the same as the previous manuscript version, it is confusing. In the first manuscript version, 0-6.0V range was quoted, and in the authors response to comment 12, 0-6V was also discussed. Please clarify the correct information (what was the actual voltage range used for the measurement the provided the data shown in Fig 6.) Fig 6 x-axis range can be kept as-is (showing the important 3-6 V range of relevance for oxidation processes). But the actual range used for measurement is critical and must be correctly stated. Please provide charts of the full data (from starting voltage) as supporting information.
-----
13. Figs 5 and 10: ideally, EIS data should be presented in square (not rectangular) graphs, with the same numerical range for x-axis and y-axis (to easily visualise any deviation from perfect semicircle)
Author response to reviewer comment 13:I'll adjust the horizontal and vertical coordinates in the EIS plot to make it easier to see any deviations from a perfect semicircle.
*Reviewer response for comment 13: Thank you, but in addition to having the same numerical range, please also make the plot area square not rectangular. (not only the numerical range, but also physical length of x and y axes should be equal).
-----
14. Line 268: "proved" Please don't use "proof"/"prove"/"proved", these are absolute expressions which should not be used in the context used here. The authors intended meaning would be better represented by "This supports the proposal that the coated separator...", or "This demonstrates that the coated separator..." or similar phrasing.
Author response to reviewer comment 14:I have checked the phrasing in the manuscript to tighten it up.
*Reviewer response for comment 14: Thank you.
-----
*15. In the discussion about ionic conductivity / pores / amount of absorbed electrolyte (lines 227 to 240) , the authors comment on how a different amount of electrolyte can effect ionic conductivity. However, please clarify for the readers, how was the ionic conductivity measurement performed - with separator first soaked in the electrolyte, wiped and then placed in the cell, or was constant volume of electrolyte added during the cell construction (regardless of which separator)?
-----
*16. The discussion (lines 258 to 276) about interfacial resistance is very interesting, and these can be useful findings for the scientific community. Since these cells are Li/separator/Li, I expect that the interfacial resistance varies from cell to cell, even for the same separator. Can the authors please include some comment for the readers regarding the reproducibility of the observed resistances?
-----
*17. The authors use of the words "steady-state" is not the common use for these words in electrochemistry, I would like to respectfully ask the authors to
(i) change "steady state voltage" in line 15 to "oxidative voltage limit"
(ii) change "steady-state voltage" in line 17 to "oxidative voltage limit"
(ii) change "steady state voltage" in line 20 to "oxidative voltage limit"
(iv) change "steady state current" in line 282 to "onset of oxidation current"
(v) delete "steady-state" from line 284 (i.e. change "steady-state current" to "current")
(vi) change "steady-state current" in line 287 to "onset of oxidation current"
(vii) change "steady-state voltage" in line 290 to "oxidative voltage limit"
(vii) change "steady-state current" in line 293 to "onset of oxidation current"
(xi) change "steady-state voltage" in line 385 to "oxidative voltage limit"
(x) change "steady state voltage" in line 405 to "oxidative voltage limit"
-----
*18. The authors show in the LSV data that the three samples, W-PE, W-PE-A and W-PE-B, have stability up to 4.2, 4.7 and 5.1 V respectively. However, looking at the data, W-PE-B already shows significant current at 4.7 V (the same current as W-PE-A). I thin k it is not reasonable to claim that W-PE-B has a stability of 5.1 V based on these data. How did the authors define the stability limit? One commonly used method is to define a certain current (the oxidative limit is reached when the current exceeds e.g. 5microA), but this doesn't appear to be the method used by the authors. Please clarify for the readers.
-----
*19. The cycling data in Fig 7(b) show a dramatic drop between cycle 20 and cycle 40, especially for W-PE-B-afterpunching and W-PE-A-afterpunching. Can the authors give some comment to the readers about this and the possible reason? Can the authors also comment to the readers on the reproducibility of these cycling data?
-----
*20. The cycling data in Fig8(c2) shows an interesting shoulder-like feature for the solid black line (0.5C discharge) and solid red line (1.0C discharge), which is not observed for any of the other samples at these C rates. Can the authors give some comment to the readers about this and the possible reason?
Author Response
Dear Editor and Reviewers,
We would like to thank you for giving us a chance to revise (“Aging behavior of polyethylene and ceramics coated separator under the simulated lithium-ion battery service compression and temperature field”), and also thank the reviewers for giving us constructive suggestions which would help to improve the quality of the manuscript.
We have carefully checked the manuscript and modified it according to the comments. In the main text, changes have been highlighted. The point-to-point responses to the comments are as follows:
REVIEWER 1:Thank you to the authors for the responses. I've provided some further comments to each response (1-14) below. In addition, following the clarifications about the analyses provided by the authors, I was able to understand the later part of the paper (porosity and electrochemical measurements) and have made some further (new) comments (15-20) related to those. I would be delighted to see the authors' further responses.
- The electrolyte being used is little unclear. In the abstract (line 18) and also on line 109, "vinyl carbonate" is mentioned, but at many other points in the manuscript, EC/DEC is mentioned. It is also rather unclear as to which experiments use 1M LiPF6 in mixed solvent, and which use just mixed solvent (i.e. no salt). e.g. line 118, line 141, and the reasons for this.
Author response to comment 1: Vinyl carbonate is abbreviated as EC and diethyl carbonate is abbreviated as DEC, which have been standardized as EC and DEC in the manuscript. The liquid electrolyte of lithium-ion battery includes lithium salt and solvent, the lithium salt is LiPF6, which plays the role of providing active lithium ions after dissolved in the solvent, in the cyclic compression process, taking into account the environmental safety, so only the synergistic effect of the solvent field is taken into account, using a mixed solvent of EC/DEC, and the other tests are all using 1M LiPF6.
Reviewer response for comment 1: I checked the manuscript and noted that there is no longer any mention of "vinyl carbonate" (which might be easily confused with vinylene carbonate, not the same as ethylene carbonate). The changes for comment 1 are appropriate and complete. Thank you.
- In the caption of Fig 2, is mentioned that the surface particles of the cyclic compression-treated separators with coating were washed away before SEM images were recorded. It is not clear if this washing process was also carried out before other experiments with these separators e.g. pore size distribution, electrochemical tests etc. I would like to ask the authors to make this clear. The presence or absence of the residual particles from the surface coating will make a great difference to the results, and so it is difficult to make further comment on the subsequent data without this being clarified.
Author response to comment 2: No cleaning of the particles was carried out for the other experiments, except that the samples were cleaned before taking the SEM images. The aim of this experiment was to find out whether the ageing behaviour of the coated modified separator would be better in terms of retaining certain electrochemical properties compared to the uncoated pure polyethylene separator. Secondly, the unavoidable plugging of the holes by a few particles was analyzed to see if it would directly cause the battery to fail, thus determining that coating the separator with oxide particles would not only help to increase the discharge capacity, but also slow down the ageing behaviour and increase the number of charge/discharge cycles to a certain extent.
*Reviewer response for comment 2: Thank you. For the readers to understand better, please add the following after "... under vacuum." on line 124.
"For the coated samples, W-PE-A and W-PE-B, after cyclic compression the surface particles were washed away prior to SEM measurement. For all other measurements after the cyclic compression, this washing was not performed."
Author response to comment 2:I've added in the manuscript.
- In Fig 2, the SEM images for W-PE-A and W-PE-B before cyclic compression, and also after cyclic compression (and removal of particles) are shown. Is it possible to see these separators with the coating removed, but without cycic compression? This would clarify to what extent the observed morphology in b2 and b3 are due to the cyclic compression, rather than pre-existing due to the coating process. I also noticed that b2 and b3 are not really described in the text. The scale bar for b3 is also missing.
Author response to comment 3:Because the substrate of the coated separator is W-PE membrane, the microporous structure of the W-PE membrane is shown after removing the coating, which is similar to the SEM image of the W-PE membrane, so it is not shown in the SEM image. a2 and a3 shows the surface topography of the coated separator without cyclic compression, and the particle morphology is obvious. I will enhance the description of b2 and b3 in the manuscript.
*Reviewer response for comment 3: Thank you. Sorry if I was not clear - I am wondering if the coating process itself has some effect on the surface morphology of the film. I think it is necessary to understand this before ascribing the morphology observed in (Fig 2) b2 and b3 to being solely due to the cyclic compression. Is it possible to show the readers some SEM images of the pristine coated separators, with coating removed? The scale bar for (Fig 2) b3 is also missing, please include a scale bar for b3.
Author response to comment 3: Figures 2 (b1),(c1) in the manuscript show SEM photographs of W-PE-A,W-PE-B without cyclic compression (corresponding to the coated surface). The following Fig.1 gives the SEM photographs of the uncoated surface for W-PE-A(left) and W-PE-B (right) without cyclic compression. Since the SEM image in Fig. 2(b3) was taken at a magnification of 10K, the original image was enlarged for the sake of uniform scale. The original image is shown as the following Fig 2. Now the scale is unified and all the photographs in the manuscript are modified.
Fig. 1 SEM photographs of the uncoated surface for W-PE-A(left) and W-PE-B (right) without cyclic compression
Fig.2 The original image of Fig. 2(b3)
- Pore size distribution is mentioned in the text around line 192 (and Fig. 3), but I couldn't find any experimental description (in the Experimental section) for how these data were obtained. Please describe in full. If possible, can data for W-PE-A and W-PE-B before cyclic compression (punching) be included?
Author response to comment 4:I have added a specific description in the experimental section and other reviewers have mentioned the same issue. In addition, to address the issue of the data of W-PE-A and W-PE-B before cycling cycle, since the gap of the coating particles will be larger than the pore size of the microporous membrane when the computer analysis is performed by this test method, it will lead to an inaccurate and not very meaningful measurement of the pore size distribution. On the other hand, since both coated films use the same W-PE film as the substrate, the measured pore size distribution after cleaning the coating particles is the same as that of the W-PE film, so it was not tested.
*Reviewer response for comment 4: On the first point, I'm sorry, I could not find the description in the Experimental section. Please add the full experimental description for the pore size distribution measurement to the Experimental section. On the second point, I understand the authors comment, but I am still wondering whether the coating process itself causes some change in the pore structure (see also my response to comment 3 above). If possible, it is better to demonstrate to the readers, with data, the authors suggestion (in their response above) that "the measured pore size distribution after cleaning the coating particles is the same as that of the W-PE film".
Author response to comment 4:Regarding the first point, the method to the obtain the pore size distribution curves has been added in the manuscript. Regarding the second point, the coating itself causes a change in porosity because the oxide ceramic particles cover the surface of the separator, and with the nano measure test programme only the pore on the surface was counted, so there is no practical significance in comparing it with the W-PE separator, because the gaps between the ceramic particles would be larger than the microporous pore size of the separator. On the other hand, since the base membrane chosen for W-PE-A and W-PE-B separators is same to W-PE, the internal pore size distribution should be the same.
- Line 107: Is the LFP cathode purchased as-is? What is the binder, conductive carbon used? Is the current collector Al, or carbon-coated Al? Is the electrode calendared (rolled/pressed)? If so, what are the conditions?
Author response to reviewer comment 5: lithium iron phosphate cathode (ratio of active substance: 91.5%)is purchased from Guangdong Canrd New Energy Technology Co. Ltd., China.Specific information has been added to the material section with specific instructions.
*Reviewer response for comment 5: Thank you. I assume the % is wt%. If so, please amend "91.5%" to "91.5wt%" in line 102.
Author response to comment 5 : I have made changes in the manuscript.
- Line 107: What are "lithium flakes"? Is it lithium sheet?
Author response to reviewer comment 6: :I've made changes in the manuscript.
*Reviewer response for comment 6: Thank you. I found the appropriate amendment in line 103.
- Line 114: what material is the hemispherical indenter? PTFE?
Author response to reviewer comment 7: The material of hemispherical indenter is customised stainless steel.
*Reviewer response for comment 7: Thank you, but please inform the readers. Please add "(customised stainless steel)" after "indenter" in line 109.
Author response to reviewer comment 7: I've made changes in the manuscript.
- Line 130: What is the contact angle measurement temperature? Were the separators soaked by full immersion in the electrolyte, or just placed on the surface of the liquid? If on the surface, for the coated separators was the coating on the upper (not in contact with liquid) or lower (in contact with liquid) face?
Author response to reviewer comment 8:The contact angle measurement is a test performed at room temperature. The angular measurement value of the contact angle is measured by dropping the electrolyte onto the surface of the separator using a syringe for electrolyte titration, and by capturing the angle formed by the electrolyte drop onto the surface of the separator, the affinity with the electrolyte is judged.
*Reviewer response for comment 8: Thank you, but please inform the readers. Please add "at room temperature" after "tested" in line 125.
Author response to reviewer comment 8: I've made changes in the manuscript.
- Equation 1 and Table 1: The % electrolyte uptake reflects the change in mass, but to fully understand, the reader should have an idea of the initial mass. Please include the average wdryvalues in the table.
Author response to reviewer comment 9: The initial mass is calculated by cutting separators of the same size and calculating the absorption rate after several tests, and then averaging the values to be more convincing, rather than calculating by averaging the masses.
*Reviewer response for comment 9: Thank you for your response. If I'd understood correctly, for each piece of separator, you weigh to obtain a wdry value, then perform immersion, then weight to obtain wwet. You then calculate Electrolyte uptake (%) according to equation 1. You perform this process at least 10 times, and take an average of the Electrolyte uptake (%) values (and so you do not calculate the average wdry or wwet values). Did I understand correctly? It is not so clear from the experimental description, could you please amend. Were the diameter of separator pieces constant? (If so, please state the diameter). In that case, please also provide the typical range of values for wdry for each of W-PE, W-PE-A and W-PE-B.
By the way, there is a typo in the text, "mersed in" has slipped down into line 142, but actually belongs in line 141 ("...was imthe liquid..." should be "was immersed in the liquid")
Author response to reviewer comment 9: I've made changes in the manuscript.
- Line 148-170: The electrolyte is not mentioned (what electrolyte and what volume). The reader should also be informed of the diameter of the electrodes and separator.
Author response to reviewer comment 10: Among them, due to the selection of CR2032 battery shell assembly, the diameter of the lithium sheet is 16mm, the diameter of the separator is 20mm, the diameter of the steel sheet is 18mm, the liquid electrolyte (1M LiPF6 in EC/DEC, volume ratio 1:1).
*Reviewer response for comment 10: Thank you, I found the appropriate amendment for the electrochemical measurement description in lines 157-160. Please see also my response for comment 9 above (regarding diameter for separator used for immersion test).
Author response to reviewer comment 10: I've made changes in the manuscript.
- Line 150: What is the voltage perturbation used for the EIS?
Author response to reviewer comment 11: Disturbance voltage amplitude is 5mV.
*Reviewer response for comment 10: Thank you, I found the appropriate amendment in the electrochemical measurement description in line 154.
- Line 159: Why is the LSV starting at 0 V? The assembled cell with SS working electrode and Li counter/reference will typically have an open circuit voltage of very roughly 2 V. Stepping this down to 0 V will result in unknown reductive processes at the working electrode, and the effect of these reductive processes (leading to products in solution and/or on the electrode surface) on the subsequent positive voltage scan is unknown. A more typical protocol for this type of LSV would be to start directly at the open circuit voltage, or initially step to a voltage slightly higher than the open circuit voltage. The voltage range of interest is likely 3 to 6 V, so it is not clear to me why it is necessary to start from 0 V, but if the authors have a different idea I would be happy to hear it.
Author response to reviewer comment 12: I didn't carefully consider the reduction process of the working electrode, and when I was looking up test methods earlier there was literature on tests done in the voltage interval of 0-6V, with the intention of seeing a smoother LSV curve.
*Reviewer response for comment 12: Thank you. However, I note that the experimental description now reads "voltage range of 2.0 to 6.0 V". Since the data shown in Fig 6 seem to be exactly the same as the previous manuscript version, it is confusing. In the first manuscript version, 0-6.0V range was quoted, and in the authors response to comment 12, 0-6V was also discussed. Please clarify the correct information (what was the actual voltage range used for the measurement the provided the data shown in Fig 6.) Fig 6 x-axis range can be kept as-is (showing the important 3-6 V range of relevance for oxidation processes). But the actual range used for measurement is critical and must be correctly stated. Please provide charts of the full data (from starting voltage) as supporting information.
Author response to reviewer comment 12:The actual measured voltage range is 0-6V, and the original curves are shown below.
- Figs 5 and 10: ideally, EIS data should be presented in square (not rectangular) graphs, with the same numerical range for x-axis and y-axis (to easily visualise any deviation from perfect semicircle)
Author response to reviewer comment 13:I'll adjust the horizontal and vertical coordinates in the EIS plot to make it easier to see any deviations from a perfect semicircle.
*Reviewer response for comment 13: Thank you, but in addition to having the same numerical range, please also make the plot area square not rectangular. (not only the numerical range, but
Author response to reviewer comment 13: Not all EIS tests show the full circular area, the properties of each material and the frequency of the test will have an effect on the EIS curve. In this mode of testing, the data is displayed as shown in the figure 5. Also, the physical meaning of the complete half circle and the present partial circle is the same, and, in fact, the impedance spectra at frequencies above 10-50 kHz mainly reflect the structure of the electrical double-layer capacitor.
- Line 268: "proved" Please don't use "proof"/"prove"/"proved", these are absolute expressions which should not be used in the context used here. The authors intended meaning would be better represented by "This supports the proposal that the coated separator...", or "This demonstrates that the coated separator..." or similar phrasing.
Author response to reviewer comment 14:I have modified in the manuscript.
*Reviewer response for comment 14: Thank you.
*15. In the discussion about ionic conductivity / pores / amount of absorbed electrolyte (lines 227 to 240) , the authors comment on how a different amount of electrolyte can effect ionic conductivity. However, please clarify for the readers, how was the ionic conductivity measurement performed - with separator first soaked in the electrolyte, wiped and then placed in the cell, or was constant volume of electrolyte added during the cell construction (regardless of which separator)?
Author response to reviewer comment 15:For liquid lithium-ion batteries, ionic conductivity is tested by adding a constant volume of electrolyte during cell construction, and it must be ensured that the electrolyte fully saturates the separator. In addition, subsequent electrochemical tests are performed with a constant volume of liquid electrolyte during dropwise addition to ensure that there is good contact between the separator and the electrodes.
*16. The discussion (lines 258 to 276) about interfacial resistance is very interesting, and these can be useful findings for the scientific community. Since these cells are Li/separator/Li, I expect that the interfacial resistance varies from cell to cell, even for the same separator. Can the authors please include some comment for the readers regarding the reproducibility of the observed resistances?
Author response to reviewer comment 16:As mentioned by the reviewer, even for the same separator, the interfacial resistance of the assembled cell may be different. Since the tests in the manuscript were carried out under the same conditions, and the tested results were used to compare the changes after cyclic compression, the values in this manuscript are acceptable.
*17. The authors use of the words "steady-state" is not the common use for these words in electrochemistry, I would like to respectfully ask the authors to
(i) change "steady state voltage" in line 15 to "oxidative voltage limit"
(ii) change "steady-state voltage" in line 17 to "oxidative voltage limit"
(ii) change "steady state voltage" in line 20 to "oxidative voltage limit"
(iv) change "steady state current" in line 282 to "onset of oxidation current"
(v) delete "steady-state" from line 284 (i.e. change "steady-state current" to "current")
(vi) change "steady-state current" in line 287 to "onset of oxidation current"
(vii) change "steady-state voltage" in line 290 to "oxidative voltage limit"
(vii) change "steady-state current" in line 293 to "onset of oxidation current"
(xi) change "steady-state voltage" in line 385 to "oxidative voltage limit"
(x) change "steady state voltage" in line 405 to "oxidative voltage limit"
Author response to reviewer comment 17: I've made changes in the manuscript.
*18. The authors show in the LSV data that the three samples, W-PE, W-PE-A and W-PE-B, have stability up to 4.2, 4.7 and 5.1 V respectively. However, looking at the data, W-PE-B already shows significant current at 4.7 V (the same current as W-PE-A). I thin k it is not reasonable to claim that W-PE-B has a stability of 5.1 V based on these data. How did the authors define the stability limit? One commonly used method is to define a certain current (the oxidative limit is reached when the current exceeds e.g. 5microA), but this doesn't appear to be the method used by the authors. Please clarify for the readers.
Author response to reviewer comment 19: When the electrode potential changes within the range of potentials where no electrode reaction occurs, there is no reduction and consumption of substances, and the Faraday current generated is 0, a smooth section. When the electrode potential exceeds the critical potential at which the substance reacts, the reduction reaction occurs and a reduction current is generated. Since the initial concentration of the substance on the electrode and surface is large at this time, there is sufficient supply of the substance to be consumed, so the reduction current gradually increases with the negative shift of the electrode potential. When the reduction current gradually increases, it is the stability limit.
*19. The cycling data in Fig 7(b) show a dramatic drop between cycle 20 and cycle 40, especially for W-PE-B-after punching and W-PE-A-after punching. Can the authors give some comment to the readers about this and the possible reason? Can the authors also comment to the readers on the reproducibility of these cycling data?
Author response to reviewer comment 19: This is a decrease in capacity due to the regrowth of the SEI membrane at the positive-negative interface. There is also a decreasing trend in 7a, but it is not very prominent. This may be due to not activating the cell enough before the test cycle to form a stable SEI film when the compressed separator is used, since the surface is not smooth after compression.
*20. The cycling data in Fig8(c2) shows an interesting shoulder-like feature for the solid black line (0.5C discharge) and solid red line (1.0C discharge), which is not observed for any of the other samples at these C rates. Can the authors give some comment to the readers about this and the possible reason?
Author response to reviewer comment 20: During the discharge process, various polarization processes in the cell will have an effect on the discharge voltage. And the above mentioned regional changes are caused by concentration polarization. Concentration polarization: Due to the hysteresis of the ion diffusion process in the solution, it causes the concentration difference between the surface of the electrode and the solution body under a certain current, which generates polarization. This polarization decreases or disappears on a macroscopic second scale (seconds to tens of seconds) as the current drops. This means that ions become more difficult to shuttle through the separator, and the ion channels become inhomogeneous leading to a condition that would be likely to occur.
Thanks again for your helpful and careful suggestion for my manuscript. I am looking forward to your future suggestion for my manuscript.
Sincerely yours,
Caihong Lei
School of Materials and Energy
Guangdong University of Technology
Guangzhou China 510006

Round 3
Reviewer 2 Report
Comments and Suggestions for Authors
The revised-3 version of the paper can be accepted for publication.
Author Response
Comments and Suggestions for Authors
The revised-3 version of the paper can be accepted for publication.
Dear Editor and Reviewers,
We would like to thank you for giving us a chance to revise (“Aging behavior of polyethylene and ceramics coated separator under the simulated lithium-ion battery service compression and temperature field”).
We have carefully checked the manuscript and modified it according to the comments.
Thanks again for your helpful and careful suggestion for my manuscript.
Reviewer 5 Report
Comments and Suggestions for Authors
Thanks to the authors for the responses.
Here I will just mention some remaining issues, with reference to the comment numbers used previously. Unfortunately there are still a couple of important issues regarding the LSV data, so I have once again selected "Major Changes".
Comment 3: Thanks to the authors for providing images of the W-PE-A,W-PE-B with coating removed. From these, I can now understand that the change in surface in morphology is from the cyclic compression, and not something due to the existing coating process. I think the readers should also have this information, so I would like to ask the authors to include these two figures as supplementary information for the manuscript. The Coatings journal allows a "Supplementary Materials" file in addition to the main manuscript, so including these two figures as Figures S1 and S2 in the Supplementary Materials file, and then mentioning them briefly in the manuscript would be very useful to the readers.
Comment 9: In the manuscript, please provide the typical range of values for wdry for each of W-PE, W-PE-A and W-PE-B.
Comment 12: Thanks to the authors for providing the additional figure showing the full range of potential for the LSV measurement. As I mentioned earlier, usually such a measurement would be performed by sweeping up from the open circuit potential (which is probably around 2 V or so). As the authors have apparently first stepped the potential down from the open circuit potential to 0 V, and then started the sweep from 0 V to 6 V there may have been some reductive processes, and indeed we see what are probably the corresponding oxidations in the range of 0.5 to 3.0 volts during the subsequent positive potential sweep. These reactions have an unknown effect on the surface of the electrode, and as such it is difficult to be confident in the determination of the oxidative limit for the system for LSVs recorded in this way. I think there are two possible options (a) repeat the LSV measurements with a more reasonable starting voltage, or (b) include these full range (0 - 6 V) figures as Figure S3 in the Supplementary Materials file, and then mentioning them briefly in the manuscript, so the readers can understand fully the limitations.
Comment 18: Sorry, I think the authors misunderstood. In Figure 6(a), it is not obvious how the data of 4.2, 4.7 and 5.1 V for the onset of oxidation was determined. The magnitude of the current is very different at each of these points. Typically, a particular magnitude of current is selected as the onset current, and applied to all systems. For example, if I zoom in on the graph and select 0.05 mA as the onset current, then I get roughly 4.5, 4.7 and 4.9 as the corresponding potentials, please see the attached figure, Figure R1 (attached as a zip file). The same concern also applies to Figure 6(b). If the authors have another method they are using to determine the onset potential, please describe it, but based on what I can see in the data, the onset potentials being quoted are not reasonable, and I would like to ask the authors to revise.
Thank you.

Author Response
Dear Editor and Reviewers,
Please see the attachment.
